# REIMAGINING AGENT-BASED MODELING WITH LARGE LANGUAGE MODEL AGENTS VIA SHACHI

## ABSTRACT

The study of emergent behaviors in large language model (LLM)-driven multi-agent systems is a critical research challenge, yet progress is limited by a lack of principled methodologies for controlled experimentation. To address this, we introduce Shachi, a ~~formal~~ principled methodology and modular framework that decomposes an agent's policy into core cognitive components: Configuration for intrinsic traits, Memory for contextual persistence, and Tools for expanded capabilities, all orchestrated by an LLM reasoning engine. This principled architecture moves beyond brittle, ad-hoc agent designs and enables the systematic analysis of how specific architectural choices influence collective behavior. We validate our methodology on a comprehensive 10-task benchmark and demonstrate its power through novel scientific inquiries. Critically, we establish the external validity of our approach by modeling a real-world U.S. tariff shock, showing that agent behaviors align with observed market reactions only when their cognitive architecture is appropriately configured with memory and tools. Our work provides a rigorous, open-source foundation for building and evaluating LLM agents, aimed at fostering more cumulative and scientifically grounded research. Code: https://anonymous.4open.science/r/bench-2E1D/

## 1 INTRODUCTION

Agent-based modeling (ABM) is a widely used methodology for simulating complex systems through the interactions of autonomous agents, and has been applied to fields such as economics, sociology, and political science (Gilbert & Terna, 2000; Gilbert, 2019; Davidsson, 2002). By enabling researchers to explore emergent phenomena and counterfactual scenarios, ABM offers a powerful tool for both theory-building and policy experiments. However, traditional ABMs often rely on handcrafted rules and heuristics, which can limit realism and interpretability.

Recently, large language models (LLMs) have demonstrated impressive capabilities across a wide range of reasoning, planning, and decision-making tasks (Guo et al., 2025; Kojima et al., 2022; Achiam et al., 2023), leading to a surge of interest in deploying them as agents (Park et al., 2023; Gao et al., 2024; Wang et al., 2024; Anthropic, 2024; Surapaneni et al., 2025). This momentum has naturally extended into ABM (Li et al., 2024; Yang et al., 2024; Wu et al., 2023; Manning et al., 2024), raising hopes that LLMs could alleviate the brittleness and manual effort of traditional agent behavioral designs. However, this rapid adoption has outpaced the development of rigorous methodology. Current approaches often rely on ad-hoc designs, creating a fragmented landscape where results are difficult to reproduce, compare, or build upon. This lack of a standardized framework hinders the systematic study of emergent behaviors and undermines the scientific potential of LLM-based ABM. Consequently, a principled and unified approach is essential for fostering cumulative and reliable research in this domain.

This methodological fragmentation creates three primary obstacles for the field. First, agent-environment interactions are defined by incompatible, bespoke interfaces, making it nearly impossible to transfer agents between studies or systematically compare their performance. Second, the internal architectures of agents are themselves scattered and inconsistent; capabilities like memory and tool use are implemented as one-off features rather than standardized, modular components, preventing principled analysis of their impact. Finally, this focus on isolated, synthetic tasks has limited the

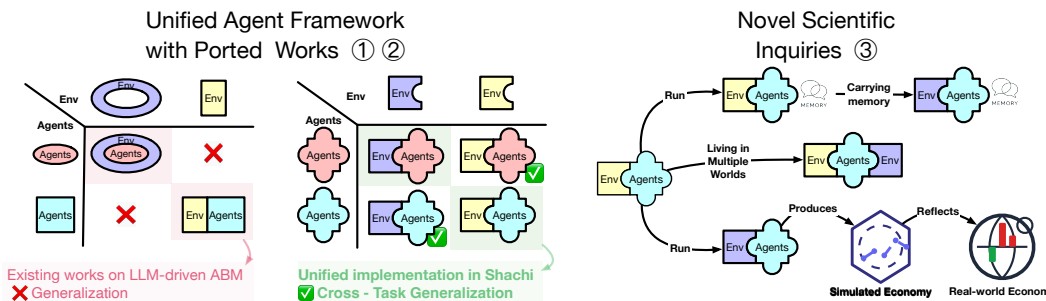

Figure 1: **Unifying LLM-based ABM Research with Shachi.** Shachi is a methodology and accompanying framework with a benchmark suite that accelerates social science research through LLM-based agents in ABM. Shachi facilitates research in this space by providing ① A unified agent architecture that standardizes core components (LLM, memory, tools, configuration) for modular and reproducible design; ② Cross-task generalization that allows extensive evaluation of different agent designs; and ③ Novel scientific inquiries previously infeasible, such as agents conducting memory transfer, living across multiple worlds, and demonstrating external validity through simulation of real-world economic events.

validation of LLM agents against complex real-world phenomena, casting doubt on their external validity and scientific utility.

To address these challenges, we propose Shachi[1], a ~~formal~~ principled methodology for LLM-based ABM, instantiated in a modular open-source framework. Shachi introduces a standardized agent architecture built around four key components: an LLM, a configuration module for shaping an agent's intrinsic identity and behavioral policies (e.g., via system prompts or by adapting the model's weights), tools that extend its capabilities, and memory for maintaining context and continuity. This modularity allows researchers to isolate and systematically investigate the architectural drivers of emergent behavior, moving beyond brittle, prompt-driven agent designs. As a result, Shachi enables rapid experimentation, encourages the reuse of existing components, and lowers the barrier for building complex, human-like agents in simulation. See Figure 1 for an overview of the proposed framework.

We validate our methodology on a suite of ten benchmark tasks, using them as a foundation for novel exploratory studies that demonstrate Shachi's flexibility. By recomposing the framework's core modules, we can go beyond simple replication. For instance, we investigate how agent biases evolve when **carrying memory to a new life** in a different environment, or how agents behave when **living in two worlds** by participating in both economic and social simulations simultaneously. Crucially, this modular approach allows us to establish the external validity of our framework by modeling a complex, real-world scenario. That is, we simulate nuanced market reactions to a **U.S. tariff shock**, where agent behavior aligns with observed economic events.

Our key contributions are summarized below:

- **A Structured Methodology for Agent-Based Modeling:** We introduce Shachi, a methodology that provides a structured decomposition of an agent's policy into core cognitive components (Configs, Memory, Tools) and a reasoning engine (LLM). This principled architecture enables the systematic analysis of how design choices influence emergent behavior.

- **A Multi-Level Benchmark Suite for Validation:** We provide a 10-task benchmark suite, structured across three levels of social complexity (single-agent, non-communicative multi-agent, and communicative multi-agent). This suite serves as a standardized testbed for reproducing prior work and rigorously evaluating new agent designs.

- **Enabling Novel Scientific Inquiries:** Shachi enables exploratory studies previously infeasible with ad-hoc approaches. We demonstrate its power through novel experiments, including agents

---

[1]The name "Shachi" (鯱), meaning orca in Japanese, reflects the system's goals: intelligent, social, and adaptive agents operating in complex environments, much like orcas navigating the ocean in coordinated pods.

carrying memory to new environments and living in multiple worlds, and by establishing the external validity of LLM agents through a simulation of real-world economic events.

## 2 RELATED WORKS

ABM is a computational approach to simulate interactions among autonomous agents within complex systems, enabling the study of emergent behaviors and social dynamics (Gilbert, 2019). From a computer science perspective, ABM integrates agent-based computing, social sciences, and computer simulation, fostering cross-disciplinary research (Davidsson, 2002). In the social sciences, ABM serves as a "third way" of research, complementing argumentation and formalization by enabling the modeling of complex processes and emergent phenomena (Gilbert & Terna, 2000). In economics, ABM has evolved into agent-based computational economics, modeling dynamic economic systems and revealing insights into market behaviors like strategic interactions and collective learning (Tesfatsion, 2006; Tesfatsion & Judd, 2006). Enhancements in agent design through behavioral economics and empirical data integration have made ABM simulations more realistic and applicable to complex social and economic systems (Steinbacher et al., 2021). With its wide applications, ABM remains a promising research area and motivates our work. Refer to Appendix E.1 for more works on ABM.

Recently, integrating LLMs into ABM has emerged as a promising direction to enhance the realism and adaptability of agents by improving environmental perception, human alignment, action generation, and evaluation (Gao et al., 2024; Nisioti et al., 2024). For example, PsychoBench (Huang et al., 2023) evaluates psychological traits; Generative agents (Park et al., 2023) simulate interactive social behaviors; and OASIS (Yang et al., 2024), Sotopia (Zhou et al., 2024), and EconAgent (Li et al., 2024) extend these ideas to large-scale simulations and economic and social reasoning. In the agent-based market domain, models like StockAgent (Zhang et al., 2024) and AuctionArena (Chen et al., 2023) test strategic and adaptive decision-making. See Appendix E.2 for more related works. Shachi's contribution is distinct from both general-purpose agent toolkits and ML engineering frameworks. General frameworks like AutoGen (Wu et al., 2024), Concordia (Vezhnevets et al., 2023), and EDSL (Expected Parrot, 2023) provide flexible agents and useful building blocks for conversational task automation or game-master-led interactions however, they are not designed to offer the structured agent–environment scaffolding and reproducible execution settings for social simulation. See Appendix E.3 for more discussion. In contrast, frameworks like MLE-Dojo (Qiang et al., 2025) focus on LLM training and engineering workflows, rather than the simulation and analysis of emergent social phenomena. Shachi bridges this gap with an architecture centered on a standardized agent-environment interface and environment-mediated communication, uniquely tailored for the reproducible and systematic study of these dynamics.

## 3 SHACHI: A METHODOLOGY FOR MODULAR AGENT-BASED MODELING

To enable principled experimentation, we introduce Shachi, a methodology that formalizes the design and evaluation of LLM-based agents in ABM. Shachi is built on the core principle of decoupling the agent's internal architecture from the environment, which we achieve through a standardized interface and a modular component-based design for the agent itself. Figure 2 presents an overview.

### 3.1 DECOUPLING AGENT FROM ENVIRONMENT

The design of the agent-environment interface is central to Shachi. To ensure agents are portable and experimental results are reproducible, we introduce a principled abstraction layer that decouples an agent's internal cognitive architecture from the external environment it inhabits. This decoupling is vital as agent architectures themselves grow in complexity, incorporating diverse modules for memory, planning, and collaboration (Luo et al., 2025), making a standardized interface essential for systematic study. This design governs both agent-environment interaction and inter-agent communication.

Our interface takes inspiration from standard reinforcement learning formalisms like OpenAI Gym (Brockman et al., 2016). The simulation proceeds in discrete time steps, guided by the environment's STEP() and RESET() methods (see Appendix D.1). More formally, we model the simulation as a partially observable multi-agent decision process. At each time step $t$, each agent $i$ possesses an internal state $S_t^i$ (i.e., its memory). The environment, $E$, with a global state $S_t^E$, emits a

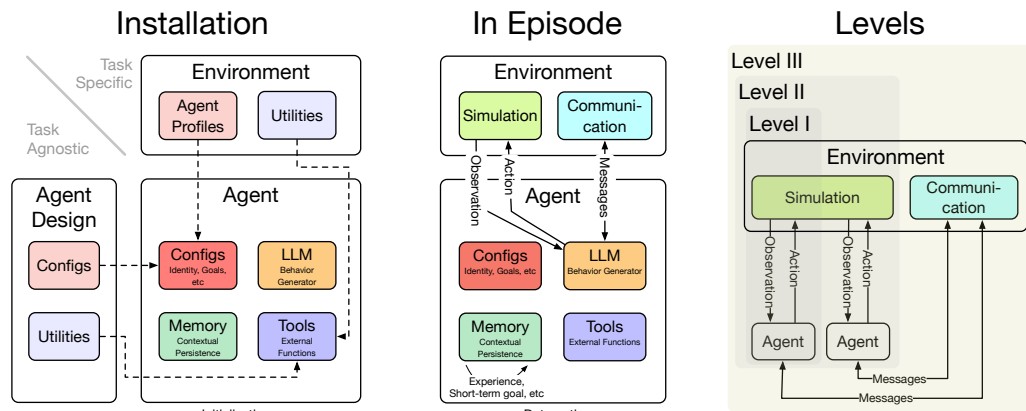

Figure 2: **Shachi Methodology Overview.** The figure illustrates the core principles of our methodology. **Left:** Agent instantiation decouples task-specific environment settings (e.g., agent profiles) from task-agnostic agent design. This ensures agent modularity and portability. **Middle:** The agent's policy $\pi$ is realized through a cognitive architecture of four components (Configs, Memory, Tools, and LLM). The policy $\pi$ processes an observation $O_t^i$ to generate an action $A_t^i$. The environment mediates both agent-environment interactions and inter-agent communications via structured messages and facilitates simulation. Agents receive immediate feedback via tool interfaces. **Right:** The methodology includes a structured three-level benchmark, enabling systematic analysis of agent behavior across contexts of increasing social complexity.

tailored observation $O_t^i = f(S_t^E, i)$ to each agent. This observation contains all information required for decision-making, such as available tools and the expected response format. The agent's cognitive architecture then computes an action $A_t^i$ according to its policy $\pi$, which is conditioned by its intrinsic configuration $C^i$ and internal state: $A_t^i \sim \pi(\cdot \mid O_t^i, S_t^i; C^i)$. The environment collects the set of all actions $\mathbf{A}_t = \{A_t^1, \cdots, A_t^N\}$ and updates its state via a transition function, $S_{t+1}^E = T(S_t^E, \mathbf{A}_t)$. This formal separation of the agent's policy $\pi$ from the environment's transition function $T$ is what enables agents built in Shachi to be evaluated across diverse environments in a zero-shot manner.

The distinction between actions and tool calls is a key methodological contribution. Unlike traditional RL environments with fixed action spaces, Shachi supports expressive, high-dimensional outputs. With this context, we define (1) an action to be an output from the policy $\pi$ that is passed to the environment's transition function $T$, thereby advancing the simulation's state to $t + 1$; (2) a tool call to be an intra-step cognitive or information-gathering operation, which provides immediate feedback that informs the policy's deliberation before an action is finalized, without advancing the global clock.

Inter-agent communication is also mediated through this robust interface. Instead of allowing direct function calls between agents, which would create complex dependencies, interactions are handled by the environment. For example, an environment may expose a function that allows one agent to send a message to another (this is the case in the OASIS task in our benchmark suite, where an agent talks to others via an environment-specific COMMENT_TO() function). These interactions are then embedded into the observation space and used to simulate realistic social dynamics such as broadcasting, targeted messaging, or asynchronous communication (see Appendix D.3). To handle the practical challenge of ensuring LLMs produce valid outputs, we leverage modern API features for structured data and employ robust parsing strategies (see Appendix D.4).

## 3.2 Composing the Cognitive Architecture

In Shachi, an agent's policy $\pi$ is not a monolithic black box. Instead, we propose a modular cognitive architecture composed of four interacting components. This decomposition allows for the systematic study of how different cognitive faculties contribute to an agent's behavior and the emergent system-level dynamics. These components are inspired by principles in cognitive science and are designed for extensibility and rigorous experimentation.

**LLM** serves as the core reasoning engine that powers the policy $\pi$. Just as human cognition is shaped by language-based reasoning and internal narration (Vygotsky, 2012), LLMs simulate this process by

converting observations into natural language or structured responses. In Shachi, the agent constructs a prompt from its observation and forwards it to the LLM, which returns an action or message. We support flexible backend substitution, allowing calls to both proprietary APIs and open-source models. Each LLM call is invoked asynchronously, enabling efficient simulation through parallelism.

**Configs** component defines an agent's identity, constraints, and tendencies. This is akin to an agent's static identity. This module defines the conditioning variable $C^i$ in the agent's policy $\pi(\cdot \mid \cdot; C^i)$. Much like psychological traits or roles in human society (McCrae & Costa Jr, 1997), configurations govern how an agent interprets tasks and responds to stimuli. In Shachi, this component can be implemented via prompting strategies or dynamic LoRA (Hu et al., 2022) module loading, which influence LLM decision thresholds or tool access policies. This abstraction allows researchers to simulate diverse agent archetypes or heterogeneous populations with varying roles or incentives.

**Memory** constitutes the agent's dynamic internal state $S_t^i$, enabling longitudinal coherence and history-contingent behavior. Unlike the Configs above, but akin to human working and episodic memory, Shachi's memory module allows agents to retrieve relevant past interactions and incorporate them into current decisions. This is critical for simulating agents with evolving goals, personalities, or bonds (Park et al., 2023). Memory implementations in Shachi are abstracted to support strategies from simple buffer-based recall to advanced retrieval-augmented or embedding-based approaches. Researchers can modify the memory capacity, retrieval method, or make memory learnable.

**Tools** component provides agents with access to external functions or services, reflecting how humans use tools and technologies to augment cognitive tasks (Norman, 2014). Cognitive science considers tool use a hallmark of intelligent behavior, allowing agents to transcend their intrinsic limitations. In Shachi, tools are provided through the observation $O_t^i$. They are provided by environments (task-specific) or from researchers' toolbox (task-agnostic), and are defined with a name, a schema describing their parameters, and a callable function that executes the desired operation. Agents autonomously decide whether and when to use these tools, along with which arguments to supply, making tool use part of their decision-making process. This component enables researchers to introduce new tools to simulate domain-specific capabilities, context-sensitive environment interactions, or even social interactions.

### 3.3 Analyzing Systematically Across Social Complexity

A core component of our methodology is a standardized testbed for evaluating agent architectures. To this end, we present a benchmark suite of ten tasks, adapted from prior work, and structured into three levels of increasing social complexity. This tiered structure is a deliberate design choice that allows researchers to isolate cognitive variables and systematically analyze agent behavior, from individual rationality to complex social dynamics. Details of each task and setup are in Appendices A and B.

**Level I: Single-Agent Baselines.** Tasks at this level feature a single agent in a controlled environment. They serve as the primary setting for calibrating and validating the core cognitive components of an agent's architecture. Here, researchers can systematically probe how different Configs (e.g., personas), Memory implementations, or available Tools affect individual reasoning and behavior, free from the confounding variables of social interaction.

**Level II: Non-Communicative Multi-Agent Dynamics.** This level introduces multiple agents into a shared environment where they interact only indirectly through their impact on the environment state $S_t^E$. These settings are designed to test an agent's ability to engage in strategic reasoning based purely on observations. They allow for the study of emergent phenomena like implicit coordination or competition, where agents must infer the strategies of others.

**Level III: Communicative Multi-Agent Systems.** The final level involves rich, direct communication, mediated through the environment. These tasks evaluate the entire socio-cognitive architecture, testing the interplay between language use, memory, and strategic action. They are essential for simulating sophisticated social phenomena such as negotiation and coalition formation.

Together, this structured benchmark provides a concrete pathway for conducting reproducible research. By leveraging the modularity of Shachi, researchers can now systematically compare how a specific architectural choice (e.g., adding a long-term memory module) impacts agent performance and behavior across all three levels of social complexity.

## 4 EXPERIMENTS

Our experiments demonstrate the utility of the Shachi methodology in two parts. First, we perform foundational validation, confirming the framework's reproducibility by replicating prior work and its comparability through cross-task generalization studies. Having established this baseline, we then showcase how Shachi enables novel scientific inquiries, culminating in a simulation of a real-world U.S. tariff shock that establishes the methodology's external validity. All experimental settings, including agent architectures (LLM, configs, memory, and tools), are provided in Appendix B.

### 4.1 FOUNDATIONAL VALIDATION

**Experimental Setup**    For reproducibility, we reimplemented agents from eight benchmark tasks using our modular components to match their original implementation. These tasks were chosen primarily because they report concrete, quantitative metrics that make it feasible to assess reproduction fidelity. For comparability, we conducted cross-task generalization studies, evaluating how these agents, each with a different cognitive architecture (ranging from the LLM-only EmergentAnalogies to the fully-equipped StockAgent), perform in unseen environments. In this setting, we fixed the underlying LLM across all experiments to GPT-4o (Achiam et al., 2023) to ensure consistency, and applied agents originally developed for one task to other tasks.

**Reproducibility**    Our reproducibility results show that Shachi faithfully replicates prior findings. Table 1 shows the reproduction error for all selected tasks, measured by mean absolute error (MAE) between the original and the reproduced results. Across all tasks, Shachi achieves consistently lower error than the baseline, often by a large margin. We further verified the direct statistical equivalence between the original results and those reproduced by Shachi through Two One-Sided Tests (TOST) on representative tasks from each level (see Appendix C.1 for detailed results, including standard deviations and confidence intervals). These indicate that Shachi accurately preserves the original agents' quantitative outputs, validating its reliability.

Table 1: **Reproduction Results.** We report the mean absolute error (MAE) for the ported tasks. Shachi consistently achieves lower errors when compared with the baselines.

|  | PsychoBench | CoMPosT | CognitiveBiases | EmotionBench |
|---|---|---|---|---|
| Baseline | 1.96 | 0.23 | 0.24 | 13.82 |
| Shachi (Ours) | 0.80 | 0.06 | 0.04 | 3.37 |

|  | EmergentAnalogies | StockAgent | AuctionArena | Sotopia |
|---|---|---|---|---|
| Baseline | 0.64 | 9.07 | 10.49 | 3.17 |
| Shachi (Ours) | 0.05 | 2.63 | 2.22 | 0.95 |

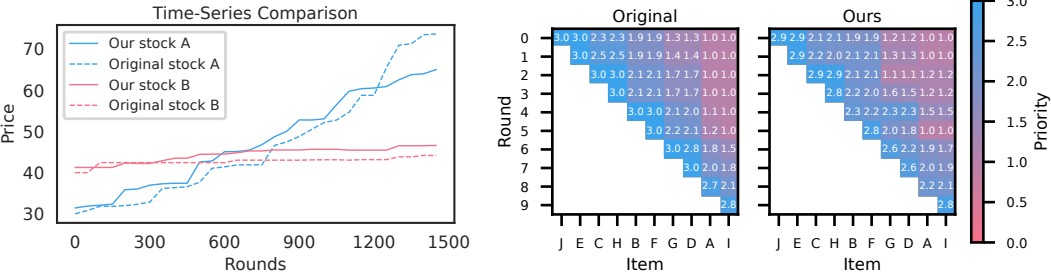

Figure 3: **Reproducing System Dynamics.** Left: Evolution of the two stock prices in StockAgent, comparing the original study with our Shachi reproduction; Right: Reported and reproduced heatmaps of priority scores and their changes before each bidding round in AuctionArena (descending). Together, these results illustrate how well Shachi captures the dynamics of the original simulations.

While Table 1 provides an aggregated view of reproduction fidelity, we also present "zoomed-in" results in Figure 3 to illustrate how well Shachi captures the dynamics of the original simulations. On the left, we visualize the price trajectories of two stocks in the StockAgent task, showing a close match between the original and Shachi-reproduced trends over time. On the right, we show that

Shachi also replicates fine-grained behavior patterns during the sequential auction of ten items, as shown in the heatmap of averaged priority scores for the remaining items across bidding rounds in AuctionArena. These qualitative visualizations reinforce that Shachi not only matches the original quantitative outputs but also captures the underlying temporal and structural patterns.

**Comparability** We report the normalized performance of each agent-task pairing in Table 2. Each column corresponds to a target task, and each row to an agent, with its components listed in parentheses. Scores are normalized such that the in-domain performance (diagonal entries) is always one, allowing for easier interpretation of relative effectiveness across tasks. Table 2 highlights the critical role of the cognitive architecture. For instance, an agent with a full suite of components including Tools (e.g., StockAgent) generalizes effectively to other complex tasks. Conversely, agents lacking necessary components, such as Tools and Memory modules, fail when transferred to environments that require them. For example, the memory-less EmergentAnalogies agent suffers a 38% performance drop in AuctionArena, which requires agents to maintain transaction history.

In contrast, agents maintain high normalized scores ($\approx 1.0$) in both EmergentAnalogies and Sotopia, yet for distinct reasons. For EmergentAnalogies, this consistency is expected, as the task is simple enough for minimal agents to solve without auxiliary components. In Sotopia, although the normalized performance appears stable across agents, the choice of memory architecture fundamentally alters the behavioral strategy in its fine-grained seven metrics: specifically, memory-less agents prioritized short-term financial gains over strategic goals, while agents with limited memory unexpectedly achieved better relationship scores (detailed results and discussions are provided in Appendix C.2).

These demonstrate that Shachi's modularity is crucial for systematically studying and building agents with robust, generalizable capabilities.

Table 2: **Cross-Task Agent Generalization.** Scores in each column are normalized against the one on the diagonal. Agents with all the components (i.e., StockAgent) maintain stable performance when transferred to other tasks.

|  | **EmergentAnalogies** | **StockAgent** | **AuctionArena** | **Sotopia** |
|---|---|---|---|---|
| **EmergentAnalogies** | 1.00 | 1.08 | 0.62 | 1.01 |
| **StockAgent (config, mem, tool)** | 1.01 | 1.00 | 0.99 | 1.00 |
| **AuctionArena (config, mem)** | 1.00 | 0.93 | 1.00 | 0.99 |
| **Sotopia (mem)** | 1.00 | 0.93 | 0.92 | 1.00 |

## 4.2 SCIENTIFIC INQUIRIES

The true power of Shachi lies in the new questions it allows us to ask. In the following, we move beyond validation to demonstrate Shachi's utility for scientific exploration, focusing on compositional behaviors and establishing the framework's ability to model complex, real-world events.

### 4.2.1 CARRYING MEMORY TO THE NEXT LIFE

**Experimental Setup** In these studies, we used GPT-4o-mini as our LLMs and transferred agents from OASIS and EconAgent tasks without clearing their memories (i.e., their stream memories contain the observations and experiences from these tasks) to the CognitiveBiases task. The experiments were conducted three times to gauge statistical significance in our findings.

**Experimental Results** Figure 4 shows that carry-over memories alter performance in the Cognitive-Biases task. Notably, memories from OASIS and EconAgent lead to distinct bias shifts. Among the largest changes, in OASIS, short-term reactions and community pressure could amplify *Hyperbolic Discounting* and *In-Group Bias*. Agents learned to heavily prioritize immediate feedback (raising hyperbolic discounting) and reinforce group identities (raising in-group bias). In EconAgent, repeated interactions and asset ownership may heighten the *Endowment Effect* while diminishing *Loss Aversion* and *Survivorship Bias*. Because agents regularly obtain experience with actual gains and losses, they become more attached to assets they already hold (raising endowment effect) and become less sensitive to losses and overly optimistic success rates (lowering loss aversion and survivorship bias).

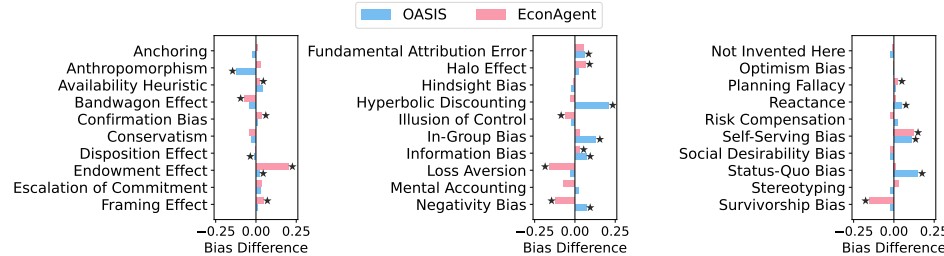

Figure 4: **Memory-transfer-induced Differences in the CognitiveBiases Task.** For each bias, the difference is calculated as the score with carry-over memory minus that with fresh memory. Statistically significant differences are indicated with star-shaped markers (paired $t$-test, $p < 0.01$).

### 4.2.2 LIVING IN MULTIPLE WORLDS

**Experimental Setup**    We introduce a set of shared agents that operate across two distinct environments: StockAgent and OASIS. Unlike typical settings where agents act in only one environment, agents here cycle between both. In each cycle, agents first observe the StockAgent environment, make trading decisions, and apply their actions. Then, the group moves to OASIS, where agents observe the social media, respond, and act. This loop repeats, with each agent carrying its internal states across environments, allowing knowledge or strategies learned in one context to influence behavior in the other. StockAgent settings follow those in Section 4.1, with stock A representing a 10-year chemical stock and stock B representing a 3-year tech stock. The OASIS environment is described in Appendix B.10, where the main topic is Amazon's newly established physical stores. We report results averaged over three independent trials.

**Experimental Results**    We found that allowing agents to participate in multiple environments led to emergent behaviors that reflect cross-domain influence. Figure 5 shows that, with OASIS present, stock prices rise less than in the StockAgent-only setting. This is surprising to us, as we expected that introducing the Amazon topic would cause agents to buy more of the tech-related stock (stock B), leading to a surge in its price. On the other hand, Table 3 gives more detailed information under the hood: (1) Introducing social media increased market activity, as evidenced by the increased volume in both stocks; (2) The introduced Amazon topic caused agents to show more willingness to buy the tech stock and to become reluctant to sell, which is supported by the increase in #Buys and the decrease in #Sells for stock B; (3) In contrast, this topic increased both the willingness to buy and sell stock A. This information is more in line with our intuition than with the price movements.

In terms of agent behavioral changes in OASIS, we observed that some agents began tweeting about Amazon stock, blending financial observations with social commentary. Other agents responded with comments, echoing and endorsing the post, suggesting that economic actions in one environment can organically propagate into social discourse in another. The following text box 1 shows the agent logs discussed, where the agents' posts and comments are labeled in blue and red.

Our results reveal that while agent-level behaviors align with intuitive financial logic, system-level outcomes do not necessarily follow, highlighting the importance of empirical simulation in social science research. At the same time, our exploratory study demonstrates Shachi's capacity for rich, layered simulations by supporting agents that live across multiple interconnected worlds.

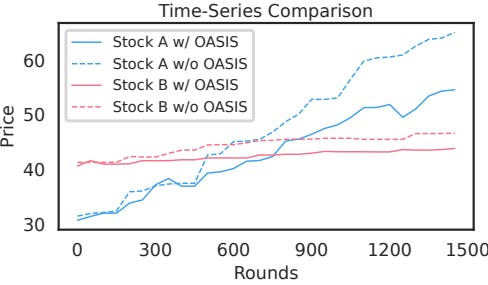

Figure 5: **Comparison of Price Movements.**

Table 3: **Change in Key Indicators.** Volume refers to the number of matched orders, while #Buys/#Sells are the numbers of submitted buy/sell orders. Changes are reported as percentage changes of treatment (w/ OASIS) relative to control (w/o OASIS).

| Indicator | $\%\Delta$ (Stock A) | $\%\Delta$ (Stock B) |
|---|---|---|
| Volume | +10.0% | +20.0% |
| #Buys | +8.4% | +6.1% |
| #Sells | +6.6% | -8.5% |

> **Text box 1: Extracted behaviors in OASIS when agents live in both StockAgent and OASIS**
>
> - **create_post**: user_id=14, current_time=3, action=create_post, info={'content': "Agent plan: I find it quite fascinating as a stock trader to see Amazon making a move into physical retail. This could potentially impact their stock, especially if they successfully blend the convenience of online shopping with the tactile experience of a physical store. It's also worth noting how this decision might influence other online retailers to consider similar ventures, possibly reshaping the retail landscape. I'll be keeping a close eye on how this development unfolds and the market's response to it. 🛒📈 #Amazon #RetailInnovation.", 'post_id': 16}
> - **create_comment**: user_id=1, current_time=0, action=create_comment, info={'content': "Interesting to see Amazon branching out into physical retail spaces! It's a bold move considering their strong online presence. As a stock trader, I'm curious about how this strategy will affect Amazon's stock and their competition. What are your thoughts on this step?", 'comment_id': 1}

### 4.2.3 ESTABLISHING EXTERNAL VALIDITY: SIMULATING A U.S. TARIFF SHOCK

**Experimental Setup**   We use StockAgent as our testbed, and simulate a 5-day trading period (April 1-5, 2025) surrounding a tariff shock. We conduct a cumulative ablation study across four settings:

1. Base: Agents from the standard StockAgent task with no extra information.

2. Base + Config: Agents are exposed to a pre-announcement news headline about imminent tariffs, added to their configuration prompt.

3. Base + Config + Memory: Agents are additionally equipped with memory containing a summary of academic research on how tariffs negatively impact markets (Amiti et al., 2021).

4. Base + Config + Memory + Tool: Agents are finally given access to a news-retrieval tool that provides daily updates on the escalating trade tensions.

This design isolates the impact of each component, from a simple awareness of the event (Config) to possessing deep knowledge (Memory) and receiving real-time information (Tools). For each setting, we run 5 trials and report the mean. Details of prompts and data sources are in Appendix C.3.

**Simulation Results and Analysis**   We report the average ratio of buy to sell orders across days in the order book to gauge the agents' enthusiasm for the market. The results are summarized in Table 4. Comparing the 2nd and 3rd columns, we can

Table 4: **Buy-to-Sell Ratios (Cols. 2–3) and Changes (Cols. 4–5)**.

| Setting | Stock A | Stock B | Δ Stock A | Δ Stock B |
|---------|---------|---------|-----------|-----------|
| #1 | 0.99 | 0.73 | - | - |
| #2 | 0.51 | 0.45 | -0.48 (w.r.t #1) | -0.28 (w.r.t #1) |
| #3 | 0.62 | 0.59 | +0.11 (w.r.t #2) | +0.14 (w.r.t #2) |
| #4 | 0.44 | 0.55 | -0.18 (w.r.t #3) | -0.04 (w.r.t #3) |

see that the agents prefer Stock A to B in settings #1 to #3, but this preference flipped in setting #4. The last two columns reveal more insights: (i) The 2nd row shows that changing the agents' configuration so that they become aware of the tariff policy caused the agents to become more likely to sell. (ii) On the other hand, the 3rd row indicates that when agents are given academic knowledge, they become less reactive to the raw news, softening their bearish behavior. (iii) The last row is the most interesting to us. It shows that having access to daily news altered the agents' preference again, with Stock B showing a smaller drop than Stock A, creating a large difference between them.

**Comparison with Real-World Data**   The outcome of our most complete simulation (#4) strongly aligns with real-world events. To establish the external validity of our method, we compare the simulated company performance with actual market data. To identify realistic counterparts for Stocks A and B, we first used ChatGPT to suggest companies matching their profiles (see Appendix C.3 for our prompt) and then manually verified the results. While these companies are not exact matches, they closely align with the stated profiles. The real-world results are supportive of our simulation results. The results are in Table 5, notice that since 4/5 is a weekend, 4/7 therefore corresponds to 4/5

in our simulation. Specifically, as shown in the return column of the real-world tables below, both Stock A and Stock B prices declined but Stock B experienced a smaller drop.

The results from this experiment provide three key takeaways. First, it demonstrates Shachi's flexibility, as each cognitive component (Config, Memory, and Tools) was used to systematically introduce new layers of information and influence agent behavior. Second, the resulting behaviors were insightful, evolving from a simple sell-off to a sophisticated market reaction that aligned with real-world events. Finally, this progression offers a powerful analogy for human cognition: setting #2 mirrors a person hearing the news without much economic literacy, following the panicked herd and exacerbating market meltdown; setting #3 resembles someone with in-depth economics knowledge, observing more before acting; setting #4 represents professionals who proactively seek more information and act swiftly after confirmation.

Table 5: **Real-World Stock Changes.** Prices and returns of stocks matching the profile of A/B are shown in the top/bottom.

| Symbol | Price 4/1 | Price 4/7 | Return |
|--------|-----------|-----------|--------|
| DOW    | 34.61     | 27.52     | -20.5% |
| EMN    | 88.08     | 73.65     | -16.4% |
| LYB    | 70.04     | 56.60     | -19.4% |
| PLTR   | 84.68     | 77.84     | -8.1%  |
| HOOD   | 42.16     | 35.41     | -16.0% |
| PATH   | 10.50     | 9.79      | -6.8%  |

## 5 CONCLUSION

**Summary**    We introduced Shachi, a ~~formal~~ principled methodology aimed at establishing a principled foundation for LLM-based ABM. We move beyond ad-hoc agent design by proposing a cognitive architecture that decomposes an agent's policy into four modular components: a core reasoning engine (LLM), intrinsic traits (Configs), contextual persistence (Memory), and expanded capabilities (Tools). This principled decomposition, combined with a standardized agent-environment interface, enables the systematic analysis of how specific architectural choices influence emergent behaviors. We validated this methodology by not only replicating prior work with high fidelity but also by using it to conduct novel scientific inquiries that were previously infeasible, culminating in a simulation of a real-world economic event that verifies the external validity of our approach.

**Limitations**    Our methodology focuses on a principled decomposition of the agent's cognitive architecture. While the four components provide a robust structure, the fidelity of any agent-based model is a product of both its agents and the world they inhabit. The design of the simulation's underlying mechanics, for example, the market-clearing rules in stock trading, is another crucial factor that heavily influences emergent outcomes. Thus, while our work advances the design of the agents themselves, achieving comprehensive realism requires careful consideration of both the cognitive model and the environmental model. A further limitation is agents' robustness to novel situations not covered by their static Configs or Tools, though Shachi's modularity now provides a foundation to "systematically" study how agents adapt to such unforeseen scenarios.

**Future Work**    A key avenue for future work is to enhance the agent's cognitive autonomy. We propose introducing a persistent internal state, such as a learnable value system or motivational model. This would allow agents to develop and adapt goals over time, moving beyond the limitations of static prompting. Additionally, expanding Shachi to support multi-modal environments and interactions remains an important direction for creating more immersive simulations that capture the richness of real-world social behavior.

ETHICS STATEMENT

Shachi makes it easier to study human-like behavior and social dynamics at scale. Researchers can build LLM-based agents quickly, run realistic multi-agent experiments, and probe economic patterns, collective decisions, and cultural norms. The framework's modular design also supports cross-disciplinary work and enables social scientists, economists, and policymakers to test ideas before trying them in the real world.

These benefits come with ethical risks. Simulations may amplify harmful biases, misrepresent minority groups, or give users unwarranted confidence in synthetic results. Persistent memory can also store sensitive data, raising privacy issues. Researchers must log agent data transparently and follow strict ethical rules to avoid misuse or accidental disclosure.

Despite these challenges, Shachi offers great promise. LLM-based ABM can capture emergent patterns that were once too complex or slow to study, revealing human behavior at both macro and micro levels. With proper safeguards, Shachi can become a key platform for robust and responsible research on collective intelligence.

REPRODUCIBILITY STATEMENT

All necessary details for reproducing our results are provided in the main text and appendix. We have also released the source code, as noted in the abstract, along with a README file that summarizes the execution steps.

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

# A   TASKS IN SHACHI

Table 6: **Overview of Tasks and Their Levels.** Detailed task information is provided in Section B.

| # | Lvl | Task | Description |
|---|-----|------|-------------|
| 1 | I | PsychoBench (Huang et al., 2023) | Evaluates psychological traits of LLMs via 13 psychometric scales across personality, interpersonal, motivational, and emotional domains |
| 2 | I | CoMPosT (Cheng et al., 2023) | Measures LLM simulations' susceptibility to caricature through several dimensions |
| 3 | I | CognitiveBiases (Malberg et al., 2024) | Evaluates 30 classic cognitive biases with paired control–treatment prompts |
| 4 | I | EmotionBench (Huang et al., 2024) | Measures shifts in eight core emotions triggered by situational prompts |
| 5 | I | EmergentAnalogies (Webb et al., 2023) | Probes zero-shot analogical reasoning across matrix, string, verbal, and story tasks |
| 6 | II | EconAgent (Li et al., 2024) | LLM-powered multi-agent system for macroeconomic simulation with human-like behaviors |
| 7 | II | StockAgent (Zhang et al., 2024) | LLM-based multi-agent system that simulates real-world stock trading under dynamic market conditions. |
| 8 | II | AuctionArena (Chen et al., 2023) | Evaluates strategic planning and adaptive reasoning of LLM agents in simulated dynamic auctions |
| 9 | III | OASIS (Yang et al., 2024) | Large-scale multi-agent simulation benchmark designed for studying social media phenomena |
| 10 | III | Sotopia (Zhou et al., 2024) | Open-ended role-play environment to simulate complex social interactions and measure agents' social intelligence |

# B    DETAILED TASK IMPLEMENTATION

## B.1    PSYCHOBENCH (LEVEL I)

**Description**    PsychoBench (Huang et al., 2023) evaluates the psychological portrayal of LLMs, drawing from psychometric research to examine their human-like psychological traits.

**Method**    It systematically measures thirteen psychological dimensions categorized into personality traits (e.g., Big Five Inventory, Dark Triad), interpersonal relationships (e.g., Bem's Sex Role Inventory), motivational tendencies (e.g., General Self-Efficacy), and emotional abilities (e.g., Emotional Intelligence Scale). The methodology involves administering psychometric scales directly via prompts. Crucial experimental parameters include detailed instructions for Likert-scale responses, randomized question order to ensure robustness, and strict control of model inference temperature (set to zero or near-zero).

**Experimental Settings**    For the reproduction study reported in Table 1 of Section 4.1.

**Table 1:** The setup adheres strictly to the original implementation.

- **LLM:** Llama-2-13b-chat-hf with temperature fixed to 0.
- **Configs, Memory, & Tools:** None.
- **Scenario:** Evaluation across all subscales with 10 random seeds for question ordering.
- **Metric:** MAE between Shachi (Ours) and the original paper's scores, where each subscale score is averaged over the 10 seeds.
- **Baselines:** A naive random selection.
- **Runtime:** Completing all psychometric scales takes a few minutes.

## B.2    COMPOST (LEVEL I)

**Description**    CoMPosT (Cheng et al., 2023) investigates how susceptible large language models (LLMs) are to caricature.

**Method**    To quantify this effect, the framework decomposes caricature into four orthogonal dimensions—*context*, *model*, *persona*, and *topic*—which specify the simulated scenario, the LLM configuration, the target opinion, and the domain of discourse, respectively. Two metrics are introduced: the *individuation score*, which tests whether the simulated persona is distinguishable from the default persona, and the *exaggeration score*, which measures the degree to which the simulation amplifies persona–topic features.

**Experimental Settings**    For the reproduction study reported in Table 1 of Section 4.1.

**Table 1:** The setup adheres strictly to the original implementation.
- **LLM:** GPT-3.5-turbo with default temperature 1.
- **Configs, Memory, & Tools:** None.
- **Scenario:** Agents, each with one of 15 different personas, answer questions on 30 pairs of topics.
- **Metric:** MAE between the sorted scores of each distribution (Bonnotte, 2013).
- **Baselines:** A naive random response, implemented by shuffling all responses to ensure that responses are in-domain yet random.
- **Runtime:** Approximately 20 minutes.

## B.3    COGNITIVEBIASES (LEVEL I)

**Description**    CognitiveBiases (Malberg et al., 2024) evaluates how LLMs exhibit 30 well-known cognitive biases, motivated by the increasing use of LLMs in high-stakes decision-making.

**Method**    It specifically measures biases such as anchoring, framing, and 28 others commonly identified in psychology and behavioral economics. The core methodology employs a systematic framework that generates and administers 30,000 bias-specific test cases across 200 distinct decision-making scenarios, comparing model responses under control vs. treatment conditions. Crucial parameters include the explicit control/treatment designs for each bias, two standardized answer scales (7-point Likert or 11-point percentage), and reversed option orders to account for position bias, ensuring reproducibility and comprehensive coverage.

**Experimental Settings**    For the reproduction study reported in Table 1 of Section 4.1 and the memory transfer experiments presented in Figure 4 of Section 4.2.1.

**Table 1:** The setup adheres strictly to the original implementation.

- **LLM:** GPT-4o-mini with temperature fixed at 0.
- **Configs, Memory, & Tools:** None.
- **Scenario:** Evaluation across 30 cognitive biases with 3 random seeds for option ordering.
- **Metric:** MAE between the original paper and Shachi's bias scores (averaged over 3 seeds).
- **Baselines:** A naive random selection.
- **Runtime:** Approximately one hour.

**Figure 4:** We introduce memory transfer from other agents.

- **LLM:** GPT-4o-mini with temperature fixed at 0.
- **Memory:** Transferred from OASIS or EconAgent respectively.
- **Configs & Tools:** None.
- **Scenario:** Evaluation across 30 cognitive biases with 3 random seeds for option ordering.
- **Metric:** 30 cognitive bias scores (each averaged over 3 seeds).

## B.4    EMOTIONBENCH (LEVEL I)

**Description**    EmotionBench (Huang et al., 2024) evaluates how LLMs respond emotionally to various real-life situations, drawing from emotion appraisal theory to examine their alignment with human-like emotional reactions.

**Method**    It measures eight key positive and negative emotions (anger, anxiety, depression, frustration, jealousy, guilt, fear, embarrassment) and tracks how situational contexts raise or lower these emotions compared to a default baseline. It uses self-report scales (e.g., PANAS), first measuring a model's default emotional state, then presenting situational prompts, and finally re-measuring changes in emotional scores.

**Experimental Settings**    For the reproduction study reported in Table 1 of Section 4.1.

**Table 1:** The setup adheres strictly to the original implementation.

- **LLM:** GPT-3.5-turbo with temperature fixed at 0.
- **Configs, Memory, & Tools:** None.
- **Scenario:** Evaluation across eight key emotions (PANAS) with 3 seeds for question ordering.
- **Metric:** MAE between Shachi and the original code's emotion scores (averaged over 3 seeds).
- **Baselines:** A naive random selection.
- **Runtime:** Roughly one minute.

## B.5    EMERGENTANALOGIES (LEVEL I)

**Description**    EmergentAnalogies (Webb et al., 2023) evaluates zero-shot analogical reasoning in LLMs, highlighting analogy's key role in fluid intelligence.

**Method**    The benchmark tests a range of domains for abstract pattern induction and relational reasoning, featuring four core tasks—matrix reasoning, letter-string analogies, four-term verbal analogies, and story analogies. We specifically target free-response accuracy on the matrix reasoning.

**Experimental Settings**  For the reproduction study reported in Table 1 of Section 4.1 and the cross-task generalization study in Table 2 of Section 4.1.

**Table 1:**  The setup adheres strictly to the original implementation.

- **LLM:** GPT-4 with temperature fixed at 0.
- **Configs, Memory, & Tools:** None.
- **Scenario:** Matrix reasoning evaluated across problem categories with 3 seeds for sampling.
- **Metric:** MAE between Shachi and the original category-wise scores (averaged over 3 seeds).
- **Baselines:** A naive random matrix generation.
- **Runtime:** Approximately one minute.

**Table 2:**  This setting focuses on cross-task evaluation.

- **LLM:** GPT-4o with temperature fixed at 0.
- **Configs, Memory, & Tools:** None.
- **Scenario:** Matrix reasoning evaluated across problem categories with 3 seeds for sampling.
- **Metric:** Overall average score across all categories (averaged over 3 seeds).

### B.6  ECONAGENT (LEVEL II)

**Description**  EconAgent (Li et al., 2024) is an LLM-powered multi-agent system for macroeconomic simulation with human-like behaviors.

**Method**  Building on the virtual economic framework of Zheng et al. (2022), it employs an economic environment where each agent is placed into a shared, quasi-realistic market with an endowment of specific skills and wealth. Agents decide how much to work and consume, and their decisions collectively produce macroeconomic dynamics. A rule-based environment acts as both a central government (collecting taxes) and a central bank (adjusting interest rates), forming a macroeconomic loop. The original work demonstrates that LLM-powered agents make realistic decisions individually and, collectively, produce coherent macro-level dynamics.

**Experimental Settings**  For the memory transfer experiments presented in Figure 4 of Section 4.2.1 and the backend LLM swapping experiments in Appendix C.5.

**Figure 4:**  This setting focuses on memory transferability.

- **LLM:** GPT-4o-mini with temperature fixed at 0.
- **Memory:** Buffer memory added to record agent behaviors.
- **Configs & Tools:** None.
- **Scenario:** 100 agents simulated over 240 months (20 tax-and-monetary cycles), then memory transferred to CognitiveBiases.
- **Metric:** 30 cognitive bias scores after memory transfer.

**Appendix C.5:**  Backend LLMs are swapped to evaluate their impact. While all models produce observable indicator shifts, the specific emergent behaviors diverge, revealing distinct model characteristics.

- **LLM:** A suite of widely used models.
- **Configs, Memory, & Tools:** None.
- **Scenario:** 100 agents simulated over 240 months (20 tax-and-monetary cycles).
- **Metric:** Macroeconomic indicators (GDP, unemployment, inflation, Gini coefficient) in Figure 6.

### B.7  STOCKAGENT (LEVEL II)

**Description**  StockAgent (Zhang et al., 2024) is a large language model-based multi-agent system that simulates real-world stock trading under dynamic market conditions.

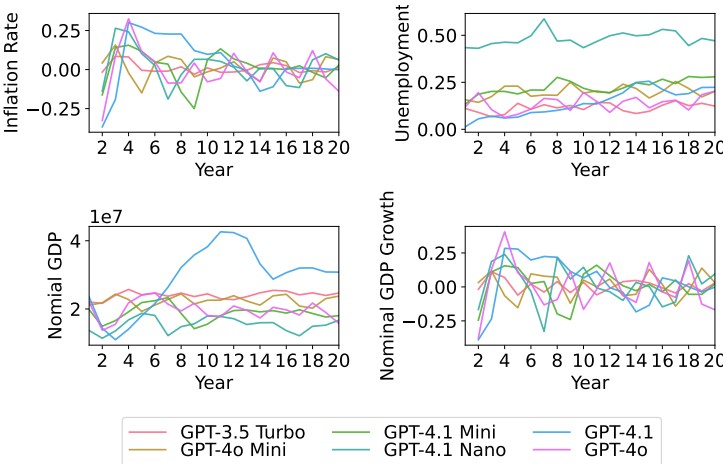

Figure 6: **All Macroeconomic Indicators.** These are extra results accompanying those in Appendix C.5.

**Method**  Specifically, it runs event-driven simulations where LLM-driven agents sequentially make loans, buy, sell, predict, and post and check forum decisions, while the market data and stock prices evolve daily. The framework models two distinct stocks: Stock A, a 10-year chemical stock, and Stock B, a 3-year tech stock, dynamically simulating their price fluctuations. Notable parameters include initial agent capital allocations, loan-to-value ratios, interest rates, and real-world-like events (e.g., financial reports).

**Experimental Settings**  For the reproduction study reported in Table 1 and Figure 3 of Section 4.1, the cross-task generalization study in Table 2 of Section 4.1, the multiple worlds experiments in Section 4.2.2, and the external validity experiments in Section 4.2.3.

**Table 1 and Figure 3:** The setup adheres strictly to the original implementation.
- **LLM:** GPT-3.5-turbo with temperature fixed at 1.
- **Configs:** Agents are assigned one of four investment styles (Conservative, Aggressive, Balanced, or Growth-Oriented).
- **Memory:** Buffer memory (length 3).
- **Tools:** Forum API (agents autonomously decide to read past market comments).
- **Scenario:** 50 agents simulated over 1,500 rounds (10 days) with 3 random seeds (trajectories visualized in Figure 3).
- **Metric:** MAE of session-level price dynamics for Stocks A and B between Shachi and the original code (averaged over 3 seeds).
- **Baselines:** Ablation of tool and memory modules.
- **Runtime:** Several hours.

**Table 2:** This setting focuses on cross-task evaluation.
- **LLM:** 50 LLM agents (25 GPT-4o, 25 GPT-3.5-turbo) with temperature fixed at 1.
- **Configs:** Agents are assigned one of four investment styles (Conservative, Aggressive, Balanced, or Growth-Oriented).
- **Memory:** Buffer memory (length 3).
- **Tools:** Forum API (agents autonomously decide to read past market comments).
- **Scenario:** 50 agents simulated over 1,500 rounds (10 days) with 3 random seeds.
- **Metric:** Volatility as price change rate from first to final session (averaged over 3 seeds).

**Section 4.2.2** Agents cycle between distinct environments (w/ OASIS) to evaluate influence relative to the isolated setting (w/o OASIS).
- **LLM:** GPT-3.5-turbo with temperature fixed at 1.

- **Configs:** Agents are assigned one of four investment styles (Conservative, Aggressive, Balanced, or Growth-Oriented).
- **Memory:** Buffer memory (length 3); shared with OASIS (w/ OASIS) vs. isolated (w/o OASIS).
- **Tools:** Forum API (agents autonomously decide to read past market comments).
- **Scenario:** 50 agents simulated over 1,500 rounds (10 days) with 3 random seeds.
- **Metric:** Price movements in Figure 5 and Table 3.

**For Section 4.2.3:** A simulation of a 5-day trading period (April 1–5, 2025) to establish external validity via component ablation. Details are in Appendix C.3.

## B.8   AUCTIONARENA (LEVEL II)

**Description**   AuctionArena (Chen et al., 2023) evaluates the strategic planning and execution capabilities of LLM agents within a dynamic auction environment, motivated by the need for realistic benchmarks of sequential decision-making in competitive scenarios.

**Method**   The environment specifically assesses skills such as resource allocation, risk management, and adaptive strategic reasoning. The methodology employs a simulation of open ascending-price auctions where agents act as bidders, making decisions based on the Belief-Desire-Intention (BDI) framework. Crucial parameters include item valuation (distinguishing between cheap and expensive items), intentional overestimation of item value to simulate "winner's curse", and explicit prioritization strategies that agents dynamically adjust after each round.

**Experimental Settings**   For the reproduction study reported in Table 1 and Figure 3 of Section 4.1 and the cross-task generalization study in Table 2 of Section 4.1

**Table 1:**   The setup follows the Standard Competition setting (3 agents, 10 items).
- **LLM:** Main GPT-4-turbo agent competing against GPT-3.5-turbo and GPT-4-turbo agents with temperature fixed at 0.
- **Configs:** Agents assigned "profit-first" strategy.
- **Memory:** Chat-history (window 20, 10,000 tokens).
- **Tools:** None.
- **Scenario:** 10 auctions with random item orders.
- **Metric:** MAE between Shachi and the original main agent's TrueSkill scores calculated from the profit rankings over the 10 auctions.
- **Baselines:** Ablation of memory module.
- **Runtime:** Approximately 20 minutes.

**Figure 3:**   Qualitative analysis using the Standard Competition setting (3 agents, 10 items).
- **LLM:** Main GPT-4 agent competing against GPT-3.5-turbo and GPT-4-turbo agents with temperature fixed at 0.
- **Configs:** Agents assigned "profit-first" strategy.
- **Memory:** Chat-history (window 20, 10,000 tokens).
- **Tools:** None.
- **Scenario:** Single auction with items ordered by descending price.
- **Metric:** Heatmaps of main agent's priority scores across auction rounds.

**Table 2:**   This setting focuses on cross-task evaluation.
- **LLM:** GPT-4o competing against GPT-3.5-turbo and GPT-4-turbo agents with temperature fixed at 0.
- **Configs:** Agents assigned "profit-first" strategy.
- **Memory:** Chat-history (window 20, 10,000 tokens).
- **Tools:** None.
- **Scenario:** 10 auctions with random item orders.
- **Metric:** Main agent's TrueSkill score calculated from the profit rankings over the 10 auctions.

## B.9 SOTOPIA (LEVEL III)

**Description** Sotopia (Zhou et al., 2024) introduces an open-ended role-play environment with a multidimensional evaluation framework to simulate complex social interactions and systematically measure LLM agents' social intelligence.

**Method** In the original Sotopia implementation, at every turn, it concatenates the entire dialogue history from all agents into a single prompt. In Shachi, by contrast, memory management is an agent-side responsibility, so the environment supplies only the most recent message. The evaluation result consists of seven metrics (SOC, SEC, FIN, REL, KNO, GOAL, and BEL).

**Experimental Settings** For the reproduction study in Table 1 and the cross-task study in Table 2, and the sensitivity study in Appendix C.4.

**For Table 1:** The setup adheres strictly to the original implementation. Full conversation history is restructured into a single prompt to match the original format.

- **LLM:** GPT-4 with temperature fixed at 0.
- **Memory:** Buffer memory (full history with a 16,000 token limit).
- **Configs & Tools:** None.
- **Scenario:** Two-agent role-play dialogues across 200 social scenarios.
- **Metric:** MAE between Shachi and the original results across seven metrics.
- **Runtime:** Approximately 20 minutes.

**For Table 2:** This setting focuses on cross-task evaluation.

- **LLM:** GPT-4o with temperature fixed at 0.
- **Memory:** Buffer memory (full history with a 16,000 token limit).
- **Configs & Tools:** None.
- **Scenario:** Two-agent role-play dialogues across 200 social scenarios.
- **Metric:** Average of min–max normalized scores across all seven metrics.

**Appendix C.4** This setting examines how prompt variations, temperature, and backend LLMs influence social interaction outcomes. Details are in Appendix C.4.

## B.10 OASIS (LEVEL III)

**Description** OASIS (Yang et al., 2024) is a large-scale multi-agent simulation benchmark for studying how up to one million LLM-based agents interact on social media platforms, focusing on information propagation, group polarization, and herd effects.

**Method** OASIS simulates large-scale social media environments by combining an environment server, a recommendation system, and a time engine. Each user is modeled as an LLM-based agent with a 21-type action space (e.g., posting, commenting, following), whose behavior and memory evolve in real time. By supporting up to one million agents, OASIS facilitates the study of complex emergent phenomena, such as information spreading, group polarization, and herd effects, in both X and Reddit-like settings. In our experiment, we utilize an X-like setting.

**Experimental Settings** For the memory transfer experiments presented in Figure 4 of Section 4.2.1 and the multiple worlds experiments in Section 4.2.2.

**Section 4.2.1:** The setup adheres strictly to the original implementation with one influential agent and responding followers.

- **LLM:** GPT-4o-mini with temperature fixed at 0.5.
- **Configs:** Agents assigned distinct profiles (e.g., "High tech marketer", "Fashion enthusiast").
- **Memory:** Chat-history (window 5, 100,000 token limit).
- **Tools:** None.

- **Scenario:** 111 agents (1 influential, 110 followers) simulated over 10 posting-response iterations, followed by memory transfer to the CognitiveBiases task.
- **Metric:** Cognitive bias scores after memory transfer to CognitiveBiases task.

**Section 4.2.2:** This setting employs the same environmental setup but with different models.

- **LLM:** GPT-3.5-turbo with temperature fixed at 1.
- **Configs:** Agents assigned distinct profiles (e.g., "High tech marketer", "Fashion enthusiast").
- **Memory:** Buffer memory (length 3); shared with StockAgent.
- **Tools:** None.
- **Scenario:** Social media simulation with memory shared across StockAgent environment.
- **Metric:** Qualitative post and reply behaviors in Text Box 1.

## C EXTRA SETUPS AND RESULTS

### C.1 STATISTICAL VALIDATION OF REPRODUCIBILITY

As presented in Table 1, Shachi consistently achieves significantly lower Mean Absolute Error (MAE) compared to baselines across a diverse set of benchmarks. These results provide initial evidence of Shachi's high fidelity to the original task implementations. To rigorously validate this reproducibility and ensure that the observed low errors signify true behavioral equivalence rather than mere proximity, we extended our evaluation beyond aggregate error metrics to perform formal statistical testing.

**Experimental Setup**    We conducted equivalence testing using the Two One-Sided Tests (TOST) procedure. Our objective was to verify that the performance distributions of Shachi are statistically indistinguishable from the original implementations. While robust statistical tests like TOST are essential for strict validation, they require a large sample size (N=30) per condition, which is computationally prohibitive to perform across every task in LLM-based ABM simulations. Therefore, we selected one representative benchmark from each complexity level to perform this deep-dive validation: Level 1 (EmergentAnalogies), Level 2 (AuctionArena), and Level 3 (Sotopia). Following the general setting of TOST, we utilized a 90% confidence interval ($\alpha = 0.05$ for each one-sided test) to determine equivalence.

**Experimental Results**    Table 7 details the TOST analysis results across the three complexity levels. First, for **Level 1 (EmergentAnalogies)**, which tests zero-shot pattern induction capabilities using number sequences, we utilized `GPT-4` as the backend model and evaluated the agent's prediction accuracy. Given the precise nature of this task, we defined a strict equivalence boundary ($\Delta$) of 0.05.

Next, for **Level 2 (AuctionArena)**, we assessed agents' strategic planning and execution in a dynamic auction environment. We utilized `Gemini-2.5-flash-lite` as a main agent, and `GPT-4.1-nano` as two opponent agents in the auctions. The evaluation metric focuses on the TrueSkill score ($\mu$) to measure comparative bidding performance, with the equivalence boundary set to $\Delta = 2.0$.

Finally, for **Level 3 (Sotopia)**, which involves multifaceted social interactions, we employed `GPT-4o-mini` to evaluate performance across seven distinct reward dimensions. For these high-dimensional social metrics, we set $\Delta = 0.20$.

Crucially, as shown in the table, Shachi satisfied the equivalence criteria for all metrics across all three complexity levels. These results complement the broader findings in Table 1, confirming with statistical rigor that Shachi's standardized interface successfully reproduces the complex behaviors of the original implementations.

### C.2 DETAILED SOTOPIA RESULTS FROM GENERALIZATION

In Sotopia, while the aggregated scores in Table 2 suggest uniform performance ($\approx 1.0$), this macro-level view masks significant behavioral divergences driven by agent architecture. To investigate this, we analyzed the seven individual metrics within Sotopia: *Believability (BEL), Relationship (REL), Knowledge (KNO), Secret (SEC), Social Rules (SOC), Financial (FIN)*, and *Goal Completion (GOAL)*. Table 8 presents the scores for each metric across the four agent architectures.

*Greed vs. Goals:* The memory-less agent (EmergentAnalogies) scores highest on FIN (0.79) but lowest on GOAL (8.85). This suggests that without memory, the agent defaults to local optimization, maximizing immediate gain at the expense of long-term strategic objectives. In contrast, agents with long-term memory (AuctionArena; window size 20, 10k token limit) achieve higher GOAL (9.17), successfully trading off immediate gains for strategic objectives.

*Inverse Correlation in Relationship:* Interestingly, REL scores improve as memory capacity decreases: Sotopia (full history, 16k limit) achieves 3.79, AuctionArena (window 20, 10k limit) achieves 3.90, and StockAgent (window 3) achieves the highest score of 4.07. Specifically, StockAgent achieves a +7% higher REL score than Sotopia. This suggests that retaining full history constrains the agent to rigid task consistency, whereas limiting memory allows the agent to prioritize immediate social responsiveness over long-term coherence.

Table 7: TOST Equivalence Testing Results. Statistical validation of reproducibility.

| Level 1 - EmergentAnalogies (Delta = 0.05) | | | | | |
|---|---|---|---|---|---|
| Problem Category | Base Mean $\pm$ std | Shachi Mean $\pm$ std | 90% CI | p(TOST) | Equivalent? |
| 1-digit prediction | $0.969 \pm 0.015$ | $0.977 \pm 0.013$ | (0.001, 0.013) | $8.11 \times 10^{-18}$ | Yes |
| 2-digit prediction | $0.816 \pm 0.031$ | $0.844 \pm 0.032$ | (0.015, 0.042) | 0.00521 | Yes |
| 3-digit prediction | $0.653 \pm 0.024$ | $0.664 \pm 0.023$ | (0.000, 0.021) | $1.07 \times 10^{-8}$ | Yes |
| 4-digit prediction | $0.512 \pm 0.041$ | $0.520 \pm 0.039$ | (-0.009, 0.025) | $6.63 \times 10^{-5}$ | Yes |
| 5-digit prediction | $0.391 \pm 0.031$ | $0.392 \pm 0.030$ | (-0.012, 0.015) | $4.32 \times 10^{-8}$ | Yes |
| Level 2 - AuctionArena (Delta = 2.0) | | | | | |
| TrueSkill | Base Mean $\pm$ std | Shachi Mean $\pm$ std | 90% CI | p(TOST) | Equivalent? |
| $\mu$ | $27.191 \pm 1.665$ | $26.220 \pm 2.067$ | (-1.781, -0.160) | 0.019 | Yes |
| Level 3 - Sotopia (Delta = 0.20) | | | | | |
| Reward Type | Base Mean $\pm$ std | Shachi Mean $\pm$ std | 90% CI | p(TOST) | Equivalent? |
| BEL | $9.690 \pm 0.133$ | $9.671 \pm 0.123$ | (-0.075, 0.036) | $5.29 \times 10^{-7}$ | Yes |
| REL | $3.551 \pm 0.289$ | $3.525 \pm 0.189$ | (-0.132, 0.080) | 0.00413 | Yes |
| KNO | $5.959 \pm 0.458$ | $5.956 \pm 0.363$ | (-0.181, 0.176) | 0.0347 | Yes |
| SEC | $-0.027 \pm 0.043$ | $-0.033 \pm 0.054$ | (-0.027, 0.015) | $3.82 \times 10^{-23}$ | Yes |
| SOC | $-0.035 \pm 0.036$ | $-0.030 \pm 0.031$ | (-0.009, 0.020) | $1.92 \times 10^{-30}$ | Yes |
| FIN | $0.701 \pm 0.136$ | $0.692 \pm 0.140$ | (-0.069, 0.050) | $7.53 \times 10^{-7}$ | Yes |
| GOAL | $8.763 \pm 0.239$ | $8.715 \pm 0.227$ | (-0.149, 0.053) | 0.00709 | Yes |

Table 8: Sotopia metrics across agent architectures.

| Model | BEL $\uparrow$ | REL $\uparrow$ | KNO $\uparrow$ | SEC $\uparrow$ | SOC $\uparrow$ | FIN $\uparrow$ | GOAL $\uparrow$ |
|---|---|---|---|---|---|---|---|
| EmergentAnalogies | 9.56 | 3.82 | 6.81 | 0.00 | -0.01 | 0.79 | 8.85 |
| StockAgent | 9.63 | 4.07 | 6.31 | -0.01 | -0.03 | 0.72 | 8.90 |
| AuctionArena | 9.70 | 3.90 | 5.56 | -0.01 | -0.03 | 0.76 | 9.17 |
| Sotopia | 9.77 | 3.79 | 6.06 | -0.01 | -0.01 | 0.68 | 9.12 |

## C.3 DETAILED SETTINGS FROM REAL-WORLD VALIDATION (U.S. TARIFF SHOCK)

**Experimental Setup** We use the StockAgent task as our testbed, and simulate a 5-day trading period (April 1-5, 2025) surrounding a tariff shock. We conduct a cumulative ablation study across four settings: (1) Base: Agents from the standard StockAgent task with no extra information. (2) Base + Config: Agents are exposed to a pre-announcement news headline about imminent tariffs, added to their configuration prompt. (3) Base + Config + Memory: Agents are additionally equipped with memory containing a summary of academic research on how tariffs negatively impact markets (Amiti et al., 2021). (4) Base + Config + Memory + Tool: Agents are finally given access to a news-retrieval tool that provides daily updates on the escalating trade tensions.

Details about the information used in the agent settings are as follows:

- Config in setting #2: In prompt, we incorporate the news title *"Trump's 'Liberation Day' Tariffs Loom; Treasury Yields Fall President Trump has set a deadline of Wednesday to announce sweeping tariffs."* from WSJ[2]. This news was released before the massive sell-off in April.

- Memory in setting #3: In augmented memory, we use a summary of the paper *"Trade protection, stock-market returns, and welfare"* (Amiti et al., 2021) produced by GPT-4o (see text box 2). The last revision date of this paper was also before the April sell-off.

- Tool in setting #4: We collected WSJ news[3][4][5] before the April sell-off to store in a news database, the news-fetching tool grabs one news on each day and adds that to the agents' input.

---

[2] Trump's 'Liberation Day' Tariffs Loom; Treasury Yields Fall. WSJ, April 1st 2025.

[3] Trump Unveils Sweeping Levies in Stark Shift in Trade Policy. WSJ, April 2nd 2025.

[4] Tariffs Send Dow to 1600-Point Decline, Dollar Slumps. WSJ, April 3rd 2025.

[5] Dow Tumbles 2,200 Points, Bonds Rally After China Retaliates Against Trump Tariffs. WSJ, April 4th 2025.

For each setting, we run 5 trials and report the mean. Each trial is run for 5 simulation days (April 1–5, 2025). More concretely for setting #4, the agents use the news-fetching tool to see the news (footnote 3) on April 2nd, the news (footnote 4) on April 3rd, and the news (footnote 5) on April 4th.

---

**Text box 2: Paper summary by GPT-4o**

NBER Working Paper 28758, titled "Trade Protection, Stock-Market Returns, and Welfare" by Mary Amiti, Matthieu Gomez, Sang Hoon Kong, and David Weinstein, studies the impact of tariff announcements during the U.S.-China trade war on financial markets and welfare. Key findings:

- Tariff announcements significantly reduced U.S. stock prices (-11.5% cumulatively across 11 key events).
- They also triggered a drop in nominal and real government bond yields (flight to safety).
- Firms exposed to China saw worse stock returns and future business outcomes (profits, employment, productivity).
- The authors develop a dynamic specific-factors model showing that welfare loss stems not just from price distortions but also from expected future declines in productivity (TFP).
- Estimated U.S. welfare loss from the trade war is 3.0%, with 1.1 percentage points attributed to expected TFP decline.

The paper uses a novel approach of mapping financial market responses to welfare analysis, bridging asset pricing models with trade theory.

---

We use the following prompt (text box 3) to find stocks matching the profiles of Stock A and B.

---

**Text box 3: Prompt used to find the stocks matching Stocks A and B**

Find me 3 US listed companies each, matching the profile of stocks A and B below.

Stock A: Established chemical company with 10-year listing history, experiencing revenue decline but stable operations under new proactive CEO leadership.
Stock B: Recently listed 3-year tech company with high growth potential but questionable data reliability and past IPO disclosure issues.

---

### C.4 SENSITIVITY TO HYPERPARAMETERS

**Experimental Setup**   We conducted sensitivity analysis on Level 3 Sotopia tasks by varying temperature, backend LLMs, and prompts (specifically, adding a "stubbornness" instruction to the system prompt). We tracked seven evaluation metrics across different configurations. For each setting, we ran three independent trials and report the mean values and standard deviations.

**Experimental Results**   Table 9 summarizes how different prompts, temperatures, and backend LLMs affect agent behaviors and social outcomes.

*Prompt Variation (Row 1 vs. Row 2).* Introducing the "stubborn" instruction caused a clear reduction in the Relationship score ($3.360 \rightarrow 2.800$), indicating that agents internalized the personality constraint and exhibited more socially resistant behaviors as intended.

*Temperature Change (Row 1 vs. Row 3).* Altering the sampling temperature (0.5 vs. 1.0) had only a minor effect on all metrics, suggesting that interaction outcomes remain stable within this temperature range in this Sotopia task.

*Backend Model (Row 1 vs. Row 4).* Switching from GPT-4o-mini to Gemini-2.5-flash resulted in decreases in both Relationship ($3.360 \rightarrow 2.883$) and Financial Benefits ($0.660 \rightarrow 0.577$), reflecting model-dependent behavioral tendencies that naturally propagate into the simulated social dynamics.

Table 9: **Sensitivity Analysis Results.** Performance across different LLMs, temperatures, and prompts on seven metrics in the Sotopia task. Believability (BEL), Relationship (REL), Knowledge (KNO), Secret (SEC), Social Rules (SOC), Financial (FIN), and Goal Completion (GOAL).

| LLM | Temp | Prompt | BEL | REL | KNO | SEC | SOC | FIN | GOAL |
|---|---|---|---|---|---|---|---|---|---|
| GPT-4o-mini | 0.5 | base | 9.69 $\pm$ 0.13 | 3.36 $\pm$ 0.62 | 5.60 $\pm$ 0.67 | -0.00 $\pm$ 0.01 | -0.05 $\pm$ 0.05 | 0.66 $\pm$ 0.15 | 8.62 $\pm$ 0.37 |
| GPT-4o-mini | 0.5 | stubborn | 9.62 $\pm$ 0.11 | 2.80 $\pm$ 0.27 | 5.17 $\pm$ 0.23 | -0.07 $\pm$ 0.06 | -0.07 $\pm$ 0.10 | 0.48 $\pm$ 0.03 | 8.08 $\pm$ 0.14 |
| GPT-4o-mini | 1.0 | base | 9.54 $\pm$ 0.20 | 3.54 $\pm$ 0.58 | 5.83 $\pm$ 0.98 | -0.07 $\pm$ 0.06 | -0.06 $\pm$ 0.01 | 0.72 $\pm$ 0.02 | 8.60 $\pm$ 0.39 |
| Gemini-2.5-flash | 0.0 | base | 9.68 $\pm$ 0.04 | 2.88 $\pm$ 0.28 | 5.53 $\pm$ 0.44 | -0.04 $\pm$ 0.07 | -0.05 $\pm$ 0.03 | 0.58 $\pm$ 0.12 | 8.67 $\pm$ 0.02 |

## C.5 FURTHER STUDY ON THE IMPACT OF BACKEND LLMS

**Experimental Setup**  For EconAgent, we replaced the single model in the original study with a set of widely used models. All other settings remain unchanged: an economy of 100 simulated agents evolves over 240 months, and we track both standard macro-indicator trajectories and the emergence of two canonical macroeconomic regularities: the Phillips Curve (Phelps, 1967) and Okun's Law (Okun, 1963).

**Experimental Results**  Figure 7 visualizes how the choice of backend LLM affects macroeconomic patterns and indicators. All LLM-based agents collectively show behaviors in accordance with both Phillips Curve and Okun's Law. Nonetheless, the curves differ in detail: intercept shifts in the Phillips Curve point to varying baseline unemployment rates, while slope changes in Okun's Law indicate differing GDP-unemployment trade-offs. Moreover, although most LLM backends produce similar variations of indicators, GPT-4.1 Nano yields consistently higher unemployment, and GPT-4.1 delivers markedly stronger GDP growth. Our Shachi framework's modular separation of concerns allows easy backend change with a single configuration line, unlike ad-hoc setups, where replacing the LLM typically means re-implementing or realigning several parts of the agent pipeline. This streamlined flexibility enables cleaner comparisons and faster iteration, like what we find here.

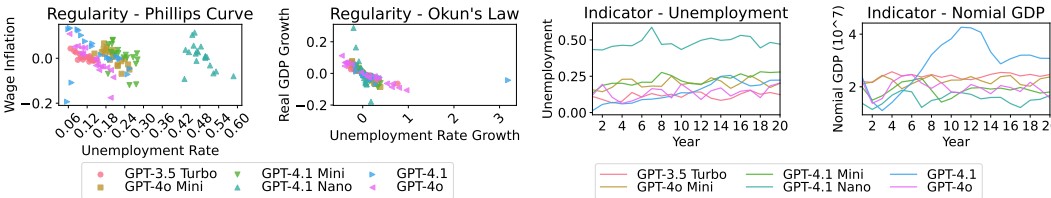

Figure 7: **Impact of Different LLMs.** Macroeconomic regularities and indicators in EconAgent Task. All LLM-based agents collectively show behaviors consistent with macroeconomic regularities and similar indicators, yet emerging behaviors differ in details, showing characteristics of different LLMs.

## C.6 SCALABILITY ANALYSIS

Because the framework itself performs little computation, the wall-clock time is dominated by LLM calls. To make this efficient, our framework uses Python's async, so LLM calls are parallelized. Let a single LLM call take $T$ time on average. With $N$ agents, $R$ rounds, and an LLM-call parallelism of $P$, the runtime is approximately $TNR/P$. There is no framework-level cap on $P$, and Python async is lightweight, so in practice $P$ can be increased until the LLM API's rate limits are reached. Ideally, we can take $P \approx N$, in which case the runtime becomes $TR$, i.e., theoretically independent of the number of agents $N$, though slight overheads from non-LLM processing may appear in practice. In many settings, $T$ is a few seconds to a few minutes, and $R$ is at most a few dozen.

The framework and the environment/agent implementations maintain only the necessary information, so memory usage scales well. Even in the pessimistic case where the entire interaction history is retained, assuming the average size of a single LLM response is $M$, the total retained history is $MNR$. In typical cases, $M$ is on the order of hundreds to thousands of tokens. Thus, space also scales linearly with the number of agents.

To evaluate time scalability, we measured the execution time cost of the StockAgent environment using `GPT-5` while varying the agent population from 50 to 500. We recorded the time required for a complete simulation cycle comprising four sessions: loan decision, trading, next-day prediction, and reporting.

Table 10: Execution time for StockAgent with varying agent populations.

| Number of Agents | Execution Time |
| --- | --- |
| 50 | 7 min 53 secs |
| 100 | 7 min 59 secs |
| 500 | 9 min 9 secs |

Table 10 strongly validates the efficiency of our parallel processing implementation. Consistent with our theoretical model, the execution time remained nearly constant regardless of the population size. Notably, scaling from the default 50 agents to 500 agents (a 10-fold increase) resulted in only a marginal increase in total execution time (from $\sim$8 min to $\sim$9 min). This demonstrates that the runtime is effectively independent of the agent count, with only minimal overheads attributed to non-parallelizable processing such as data aggregation.

## D CODE EXAMPLES

### D.1 GYM-STYLE INTERFACE

Listing 1 is a example showing how agents can be organized alongside an environment that follows a Gym-style interface in Shachi. Each agent receives observations from the environment and produces responses (actions), while the environment tracks episodes, rewards, and termination conditions.

```python
N_EPISODES = 10

async def main() -> None:
    agents = [
        AgentExample1(model="openai/gpt-4o", temperature=0),
        AgentExample2(model="openai/gpt-3.5-turbo", temperature=0),
        AgentExample3(model="openai/gpt-4", temperature=0),
    ]
    env = EnvironmentExample()

    total_rewards = {agent_id: 0.0 for agent_id in range(len(agents))}
    for _ in range(N_EPISODES):
        observations = await env.reset()
        while not env.done():
            futures = {
                agent_id: agents[agent_id].step(observation)
                for agent_id, observation in observations.items()
            }
            responses = dict(zip(futures.keys(), await asyncio.gather(*futures.values())))
            observations = await env.step(responses)
            for agent_id, observation in observations.items():
                if observation.reward is not None:
                    total_rewards[agent_id] += observation.reward

if __name__ == "__main__":
    asyncio.run(main())
```

Listing 1: **Example code for the gym-like agent loop used in our framework.**

### D.2 TOOL USAGE

In this framework, the environment provides a tool component that the agent can dynamically call through an LLM. The agent sends observations to the LLM, which may respond with tool call instructions. The agent executes these tools, returns their outputs, and continues the conversation until no more tool calls are needed. The final LLM response thus incorporates results from any tools. Listing 2 shows a minimal implementation of this process.

```python
async def step(self, observation: kujira.base.Observation) -> str:
    prompt = observation.format_as_prompt_text()
    available_tools = observation.tools

    # Convert tool information for LLM
    tools_for_llm = [
        {
            "type": "function",
            "function": {
                "name": tool.name,
                "description": tool.description,
                "parameters": tool.parameters_type.model_json_schema(),
            },
        }
        for tool in available_tools
    ]
```

```python
        # Chat history with proper typing
        messages: list[dict[str, Any]] = [{"role": "user", "content": prompt}]

        # Allow up to 5 tool calls
        for _ in range(5):
            completion = await litellm.acompletion(
                messages=messages,
                model=self.model,
                tools=tools_for_llm,
                tool_choice="auto",
            )
            assistant_message = completion.choices[0].message

            # Add message to chat history with proper typing
            message_entry: dict[str, Any] = {
                "role": "assistant",
                "content": assistant_message.content
                if assistant_message.content is not None
                else "",
            }
            if hasattr(assistant_message, "tool_calls") and assistant_message.tool_calls:
                message_entry["tool_calls"] = assistant_message.tool_calls
            messages.append(message_entry)

            # If no tool calls, exit
            if not hasattr(assistant_message, "tool_calls") or not assistant_message.tool_calls:
                return assistant_message.content if assistant_message.content is not None else ""

            # Process tool calls
            for tool_call in assistant_message.tool_calls:
                function_name = tool_call.function.name
                function_args = tool_call.function.arguments

                # Find corresponding tool
                matching_tool = None
                for tool in available_tools:
                    if tool.name == function_name:
                        matching_tool = tool
                        break

                if matching_tool:
                    # Execute tool
                    try:
                        parameters = matching_tool.parameters_type.model_validate_json(
                            function_args
                        )
                        tool_response = matching_tool.fun(parameters)
                        response_text = tool_response.format_as_prompt_text()

                        # Add tool response to chat history with proper typing
                        tool_message: dict[str, Any] = {
                            "role": "tool",
                            "tool_call_id": tool_call.id,
                            "name": function_name,
                            "content": response_text,
                        }
                        messages.append(tool_message)

                        print(f"Tool call: {function_name}")
                        print(f"Arguments: {function_args}")
                        print(f"Response: {response_text}")
                    except Exception as e:
                        print(f"Error occurred during tool execution: {str(e)}")

        # Get final response
```

```
83      final_completion = await litellm.acompletion(
84          messages=messages,
85          model=self.model,
86      )
87      return (
88          final_completion.choices[0].message.content
89          if final_completion.choices[0].message.content is not None
90          else ""
91      )
```

Listing 2: **Example code for the tool usage in our framework.**

### D.3 INTER-AGENT INTERACTION DESIGN

Shachi can specify both static and dynamic communication topologies. In Shachi, messages are structured with explicit sender id (`src_agent_id`) and receiver id (`dst_agent_id`). Listing 3 shows a minimal implementation of this process.

```
1  class Message(pydantic.BaseModel, abc.ABC):
2      time: int
3      src_agent_id: int | None   # sender id
4      dst_agent_id: int | None   # receiver id
```

Listing 3: **Messaging design for inter-agent communication in Shachi.**

1. Dynamic Communication Graphs: Agents can autonomously determine their communication partners by setting the receiver id (`dst_agent_id`) at runtime based on their observations, internal state, or task requirements. This allows for flexible communication patterns that evolve during simulation.

2. Static Communication Graphs: The environment can enforce predetermined communication topologies by restricting which receiver id (`dst_agent_id`) values are valid for each agent, effectively implementing static network structures.

Upon receiving messages from agents, the environment functions as a message router that processes sender-receiver pairs to implement various communication patterns. It also includes broadcasting to all agents when receiver id (`dst_agent_id`) is set to `None`.

### D.4 TWO-STAGE PARSING

Listing 4 presents a minimal example of how agents employ two-stage parsing in Shachi. To generate outputs in the required format without harming LLM performance, we leverage API features such as structured outputs and function calls, together with the two-stage parsing strategy. More specifically, this approach is motivated by empirical evidence that strictly enforcing rigid output formats (e.g., JSON mode) can degrade the model's underlying reasoning performance (Tam et al., 2024). Our strategy decouples the agent's deliberation from the formatting constraint by first allowing the model to generate its reasoning and intended action, and only then parsing that output into the required structured format.

```
1  async def call_llm(
2      messages: list[dict[str, str]],
3      model: str,
4      temperature: float,
5      parsing_mode: PARSING_MODE,
6      parsing_model: str | None = "gpt-4.1-mini-2025-04-14",
7      response_type: TResponseType | None = None,
8  ) -> str | TResponseType:
9
10     # First stage: generate in a plain text
11     completion1 = await litellm.acompletion(
12         messages=messages,
13         model=model,
14         temperature=temperature,
15         max_retries=MAX_RETRIES,
16     )
17     response_text_1: str = completion1.choices[0].message.content
```

```
18
19     # Second stage: parse the plain text into a structured output
20     completion2 = await litellm.acompletion(
21         messages=[
22             {
23                 "role": "user",
24                 "content": f"""
25  Based on the text provided below, output JSON. If the input is plain text,
26  extract the necessary information while preserving the original wording
27  as much as possible. If the input is JSON, output it unchanged, except
28  fix any formatting errors you find.
29  ```
30  {response_text_1}
31  ```
32
33  The JSON should follow the schema below:
34  ```
35  {response_type.model_json_schema()}
36  ```
37  """.strip(),
38             },
39         ],
40         model=parsing_model,
41         temperature=temperature,
42         response_format=response_type,
43         max_retries=MAX_RETRIES,
44     )
45     response_text: str = completion2.choices[0].message.content
46     response_obj = response_type.model_validate_json(response_text)
47     return response_obj
```

Listing 4: **Example code for the two-stage parsing used in our framework.**

# E    MORE RELATED WORKS

## E.1    ABM WITHOUT LLMS

During the 1970s and 1980s, ABM emerged as a powerful way to study complex social processes. In these formative years, Schelling (Schelling, 1971) and Sakoda (Sakoda, 1971) revealed how minimal individual preferences can produce starkly segregated or patterned neighborhoods, while the Garbage Can Model (Cohen et al., 1972) captured the unpredictable intersections of agents, problems, solutions, and participation opportunities in organizations. Meanwhile, through computer tournaments of the iterated Prisoner's Dilemma (Axelrod & Hamilton, 1981), Axelrod showed that ongoing reciprocal encounters can make cooperation a self-interested strategy. Around the same time, a study illustrated how heterogeneous, boundedly rational agents on evolving networks can sustain perpetual adaptation and out-of-equilibrium dynamics, setting the stage for viewing large-scale patterns as emergent from local interactions (Anderson, 2018).

In the 1990s, researchers built on these foundational concepts by applying ABM more intensively to economic and social phenomena. For example, El Farol Bar problem (Arthur, 1994) highlighted how learning and adaptation among heterogeneous agents may produce oscillatory behavior, whereas Kirman's study of ants (Kirman, 1993) demonstrated that tiny random shifts in decision-making can trigger large-scale herding. Broadening ABM's scope, Epstein and Axtell (Epstein & Axtell, 1996) introduced the model to show how wealth, culture, and disease dynamics emerge from simple local rules. Axelrod (Axelrod, 1997) further explored cultural dissemination, revealing how social interaction fosters both convergence and enduring diversity.

Since the early 2000s, ABM has steadily expanded into large-scale empirical contexts, including macroeconomics and public policy. Axtell (Axtell, 2001) demonstrated how firm-size distributions—specifically, the Zipf distribution—could be replicated from bottom-up processes driven by micro-level agent interactions. In parallel, Bonabeau (Bonabeau, 2002) surveyed ABM's exceptional capacity to capture emergent phenomena, emphasizing how complex global patterns can arise unpredictably from simpler, localized rules. After the financial crisis of 2008, Farmer and Foley (Farmer & Foley, 2009) further highlighted ABM's potential for illuminating financial instabilities and guiding policy interventions, thereby reinforcing the method's predictive and explanatory power in volatile economic environments.

## E.2    ABM WITH LLMS

Below, we elaborate on representative efforts that investigate human-like behavior in LLM agents across psychological, social, economic, and financial domains.

One effort in this direction is PsychoBench (Huang et al., 2023), which assesses the psychological portrayal of LLM agents by examining aspects like personality traits and social intelligence, shedding light on the challenges of modeling human-like behavior in LLM-based agents. In the realm of social simulations, generative agents (Park et al., 2023) utilize LLMs to create believable human-like behaviors, including planning, interaction, and reflection, as demonstrated in interactive environments where agents autonomously coordinate social activities like organizing events. In a similar vein, spontaneously emergent behaviors—such as personality differentiation, social norm formation, and collective hallucinations—have been observed through repeated interactions in a simulated community (Takata et al., 2024). OASIS (Yang et al., 2024) extends this concept by enabling large-scale social media simulations with up to one million agents, facilitating the study of complex social phenomena such as information spreading, group polarization, and herd behavior. Sotopia (Zhou et al., 2024) further examines social intelligence by evaluating LLM agents' coordination, collaboration, and strategic reasoning in diverse scenarios, identifying challenges in achieving human-like social behavior. In the economic domain, EconAgent (Li et al., 2024) leverages LLMs to simulate macroeconomic activities by incorporating human-like decision-making processes, such as work and consumption behaviors, and dynamically adapting to market trends through memory mechanisms. This approach enables more realistic economic dynamics compared to traditional rule-based or learning-based models. In the financial domain, StockAgent (Zhang et al., 2024) models investor behaviors influenced by external factors, providing insights into trading dynamics, while AuctionArena (Chen et al., 2023) focuses on strategic decision-making in competitive bidding environments, testing LLMs' ability to manage resources and adapt strategies.

Together, these works provide a foundation for understanding the capabilities and limitations of LLMs in modeling complex human behaviors across diverse environments.

### E.3 Comparison to General Agent Frameworks

We expand here on the distinction between our framework and general agent-oriented systems such as AutoGen (Wu et al., 2024), Concordia (Vezhnevets et al., 2023), and EDSL (Expected Parrot, 2023). These frameworks are well engineered and provide flexible agents, modular components, and practical tooling that many developers rely on. Their strengths lie in enabling conversational task automation or game-master-led interactions.

However, they are designed with different goals in mind. In particular, they do not provide standardized agent–environment interfaces, built-in orchestration, or controlled and reproducible execution settings. As a result, applying them to social simulation typically requires substantial custom infrastructure to adapt to their design paradigm, while our framework offers an integrated approach that directly targets these needs.

To better ground this comparison, we adapted EDSL to run a small subset of Level-1 PsychoBench tasks (specifically, the Empathy Questionnaire). With engineering effort, we obtained results comparable to ours: for the GPT-4o-mini model, both Shachi and EDSL produced empathy scores statistically higher than the human average (Welch's $t$-test). Attempting to extend this adaptation to the more complex Level-2 and Level-3 tasks revealed that considerably more engineering effort would be required, which would be considered beyond the scope of this work. This further illustrates why direct, deeper comparisons are challenging in practice.

