# OpenReview forum: "Reimagining Agent-based Modeling with Large Language Model Agents via Shachi"
_ICLR.cc/2026/Conference — Submitted to ICLR 2026_

### Official Review · Reviewer_qpSQ · 2025-10-28

**Soundness:** 3
**Presentation:** 3
**Contribution:** 3
**Rating:** 6
**Confidence:** 3

**Summary:**

This paper introduces Shachi, a formal methodology and modular framework designed to address the fragmentation and lack of reproducibility in LLM-based agent-based modeling (ABM). The core contribution is a principled decomposition of an agent's policy into four modular components: a reasoning engine (LLM), intrinsic traits (Configs), contextual persistence (Memory), and expanded capabilities (Tools), orchestrated through a standardized agent-environment interface. This architecture enables systematic analysis of how specific cognitive components influence emergent behaviors.

**Strengths:**

1. Shachi's core strength lies in its formal decomposition of an agent's cognitive architecture. By standardizing components (LLM, Configs, Memory, Tools) and the agent-environment interface, the framework directly addresses the field's reproducibility crisis.
2. The inclusion of a 10-task benchmark, thoughtfully structured across three levels of increasing social complexity, provides a much-needed standardized testbed for the community.

**Weaknesses:**

1. The inclusion of a 10-task benchmark, thoughtfully structured across three levels of increasing social complexity, provides a much-needed standardized testbed for the community.
2. The inclusion of a 10-task benchmark, thoughtfully structured across three levels of increasing social complexity, provides a much-needed standardized testbed for the community.


Missing reference:
Junyu Luo et al., Large language model agent: A survey on methodology, applications and challenges, arXiv 2025.

**Questions:**

Please see the weakness.

---

> ### Author Response · Authors · 2025-11-21
> **Responses to Reviewer qpSQ**
>
> We thank the reviewer for their positive assessment and for highlighting the value of our framework and the 10-task benchmark. We are encouraged that you found our standardized testbed to be a "much-needed" contribution to the community.
>
> ---
>
> Weaknesses:
> > 1, The inclusion of a 10-task benchmark, thoughtfully structured across three levels of increasing social complexity, provides a much-needed standardized testbed for the community.
> > 2, The inclusion of a 10-task benchmark, thoughtfully structured across three levels of increasing social complexity, provides a much-needed standardized testbed for the community.
>
> Thank you for your positive feedback on our framework. We would welcome any additional points for improvement or other comments you may have.
>
> ---
>
> > Missing reference:
> Junyu Luo et al., Large language model agent: A survey on methodology, applications and challenges, arXiv 2025.
>
> The Luo et al. survey reinforces our rationale by showing that agents are rapidly evolving into complex systems built from distinct functional modules like memory, planning, and tools. This trend validates our core argument: as this internal complexity increases, a principled, modular design like Shachi becomes essential for systematically studying, testing, and isolating these components. We have **added this explanation and cited the paper in Section 3.1.**

---

> > ### Comment · Reviewer_qpSQ · 2025-11-26
> >
> > Thanks for your rebuttal. I want to maintain my score since it is already a positive rating.

---

### Official Review · Reviewer_zhc3 · 2025-10-29

**Soundness:** 3
**Presentation:** 4
**Contribution:** 2
**Rating:** 4
**Confidence:** 3

**Summary:**

The paper introduces Shachi, a modular framework for LLM-based ABM that decomposes agent policy into four components (Configs, Memory, Tools, LLM). The authors claim this principled architecture enables systematic, reproducible analysis of emergent behavior, validating it with a 10-task benchmark and a real-world tariff shock simulation to demonstrate external validity.

**Strengths:**

1.	Novel Modular Framework: The paper proposes a clear, modular abstraction for agent design. This is a valuable contribution that moves the field away from ad-hoc scripts and toward a standardized, reproducible, and systematic methodology.

2.	Good Presentation: The paper is well-written, clearly structured, and easy to follow. Figures and visualizations are clear and effectively communicate the authors' core contributions and experimental results.

**Weaknesses:**

1.	Weak Theoretical Grounding: The paper presents the decomposition of (Configs, Memory, Tools, LLM) as a “formal methodology” or “principled architecture”.  In practice, this looks more like a conceptual abstraction or a software-engineering style decomposition than a framework derived through rigorous theoretical justification. The paper does not conduct a formal analysis of this, which makes the claim of a "formal methodology" seem somewhat exaggerated. But it is indeed a well-structured framework.

2.	Weak Support for Generalization Claims: In cross-task generalization research, the performance differences between StockAgent with complete components and Sotopia Agent with only Memory across tasks are very slight (0.99 vs 0.92, 0.93). The strong contextual reasoning ability of the LLM itself may be the main driving force for generalization.

3.	Weak External Validity: In the external validity experiment (Section 4.2.3), the results mainly show that Shachi’s outputs can be effectively controlled through the inputs of Memory and Tools. However, this only reflects the framework’s configurability rather than its predictive capacity. The experiment does not convincingly demonstrate that Shachi can robustly handle or generalize to real-world scenarios beyond the information explicitly provided to the agents.

**Questions:**

1.	How is “Formality” Defined in This Methodology? The paper repeatedly describes Shachi as a “formal methodology”, but the framework is primarily an implementation protocol.

2.	In Shachi, the agent is decomposed into four components—Configs, Memory, Tools, and LLM. Since both retrieved memory content and tool outputs are provided to the LLM in textual form, potential conflicts or negative interference between these inputs may occur. As a formal methodology, how does Shachi ensure that such inconsistencies or interference are systematically prevented or mitigated at the architectural level?

---

> ### Author Response · Authors · 2025-11-21
> **Responses to Reviewer zhc3 (Part 1)**
>
> We thank the reviewer for their constructive feedback and for recognizing Shachi as a novel modular framework that moves the field toward a standardized, reproducible, and systematic methodology.  We appreciate the opportunity to clarify our theoretical positioning and strengthen the interpretation of our results.
>
> To address your concerns, we have made the following revisions and additional analyses:
> - Refined Theoretical Grounding (W1 & Q1): We acknowledged the distinction between mathematical formality and structural principles. We have revised our terminology from "formal" to "principled methodology" and reinforced the cognitive science grounding in Section 3.2.
> - Expanded Generalization Analysis (W2): We provided a detailed breakdown of memory effects in non-tool contexts (e.g., AuctionArena) and analyzed qualitative trade-offs in Sotopia metrics to demonstrate the critical impact of modular components (Section 4.1 & Appendix C.2).
> - Clarified External Validity Goals (W3): We articulated that our contribution lies in the framework's configurability for explanatory power and mechanism replication, rather than pure predictive capacity.
> - Addressed Component Interaction (Q2): We explained how the framework systematically handles potential conflicts between Memory and Tool outputs via the Configs module without hard-coded restrictions.
>
> Please find our detailed responses below.
>
> ---
>
> > 1. Weak Theoretical Grounding: The paper presents the decomposition of (Configs, Memory, Tools, LLM) as a “formal methodology” or “principled architecture”. In practice, this looks more like a conceptual abstraction or a software-engineering style decomposition than a framework derived through rigorous theoretical justification. The paper does not conduct a formal analysis of this, which makes the claim of a "formal methodology" seem somewhat exaggerated. But it is indeed a well-structured framework.
>
>
>
> We thank the reviewer for their feedback and for recognizing Shachi as a "well-structured framework". Our central contribution is a principled methodology and modular framework designed to solve the "methodological fragmentation" that currently hinders reproducible and cumulative research in LLM-based ABM. Our method's novelty is detailed as follows:
> 1. **Theoretical grounding**: Our decomposition is not ad-hoc, it is explicitly grounded in established cognitive science principles. Specifically, we map Configs to psychological traits, Memory to working and episodic memory, Tools to cognitive augmentation, and cited papers for each of these in Sec 3.2.
> 2. **Empirical justification**: Our "well-structured framework" is justified by its empirical and scientific value. Our Sec 4.2.3 (US tariff shock) serves as a direct cumulative ablation study. The key finding is that only the full cognitive architecture produced agent behaviors that aligned with real-world observations. This empirically demonstrates that our decomposition is necessary for establishing the external validity of agents, proving its scientific utility beyond just good engineering.
>
> On the other hand, the reviewer makes a fair point that our claim of a "formal methodology" may be an exaggeration; we have **edited our language to be more precise by using "principled methodology" in the revision**.

---

> ### Author Response · Authors · 2025-11-21
> **Responses to Reviewer zhc3 (Part 2)**
>
> > 2. Weak Support for Generalization Claims: In cross-task generalization research, the performance differences between StockAgent with complete components and Sotopia Agent with only Memory across tasks are very slight (0.99 vs 0.92, 0.93). The strong contextual reasoning ability of the LLM itself may be the main driving force for generalization.
>
> We appreciate the observation regarding Table 2. Regarding the performance gaps (0.99 vs. 0.92/0.93), we first wish to emphasize that these figures represent a decline of approximately 6–7%. We consider this a meaningful degradation rather than a slight variation. However, we acknowledge that the significance of these margins may have been obscured in the current presentation. Thank you for giving us the opportunity to clarify the specific impact of the modular components.
>
> 1. **Importance of Memory in Non-Tool Contexts**: While the reviewer noted that differences seem limited to tool-requiring tasks, our results on AuctionArena suggest otherwise. AuctionArena does not require external tools, yet the memory-less EmergentAnalogies agent scores significantly lower (0.62) compared to memory-equipped agents (~1.0). This demonstrates that for complex tasks, the memory module is a critical performance factor, independent of tool availability.
> 2. **Qualitative Shift in Sotopia**: We recognize that the normalization in our main paper may have obscured fine-grained differences. However, examining Sotopia’s seven metrics individually reveals distinct agent profiles.
>     - **Greed vs. Goals**: The memory-less agent (EmergentAnalogies) scores highest on Financial (1.159) but low on Goal Completion (0.970). This suggests that without memory, the agent defaults to local optimization (maximizing immediate gain). In contrast, agents with long-term memory (AuctionArena; window 20, 10k token limit) achieve higher Goal Completion (1.005), successfully trading off immediate gains for strategic objectives.
>     - **Inverse Correlation in Relationship**: Interestingly, Relationship scores improve as memory capacity decreases: Sotopia (all history, 16k limit) < AuctionArena (window 20, 10k limit) < StockAgent (window 3). Specifically, StockAgent achieves a +7% higher score than Sotopia. This suggests that retaining full history constrains the agent to rigid task consistency, whereas limiting memory allows the agent to prioritize immediate social responsiveness over long-term coherence.
>
>
> These results demonstrate that both the availability and specific configuration of modules critically impact task performance. We have incorporated the analysis of memory effects (Point 1) and added the detailed Sotopia results and discussion (Point 2) to **Section 4.1 and Appendix C.2**.
>
>
>
> ---
>
>
>
> > 3. Weak External Validity: In the external validity experiment (Section 4.2.3), the results mainly show that Shachi’s outputs can be effectively controlled through the inputs of Memory and Tools. However, this only reflects the framework’s configurability rather than its predictive capacity. The experiment does not convincingly demonstrate that Shachi can robustly handle or generalize to real-world scenarios beyond the information explicitly provided to the agents.
>
>
> We thank the reviewer for this comment, as it allows us to clarify the goal of the external validity experiment.
>
> 1. **Our goal is not predictive capacity**: Predictive capacity is not our objective, as our contribution is a methodology, not a specific predictive model.
> 2. **Our goal of Section 4.2.3**: Through the experiments in Section 4.2.3, we wish to (1) demonstrate Shachi's flexibility and how our modular design allows for the systematic cumulative study (which is not possible in prior works), and (2) establish external validity by showing that this principled 4-part architecture is necessary to replicate a complex, observed real-world event.
> 3. **Configurability is the scientific contribution**: We thank the reviewer for acknowledging Shachi's configurability, as this is precisely the scientific contribution. Prior ad-hoc models lack this principled configurability, preventing them from systematic analysis and extension. It is only when we use Shachi's principled architecture to systematically add cognitive components that the collective agent behavior aligned with the observed market reaction.

---

> ### Author Response · Authors · 2025-11-21
> **Responses to Reviewer zhc3 (Part 3)**
>
> > 1. How is “Formality” Defined in This Methodology? The paper repeatedly describes Shachi as a “formal methodology”, but the framework is primarily an implementation protocol.
>
>
> We thank the reviewer for this feedback, which rightly points out the ambiguity of the term "formal methodology". In the revised version of our paper, we have edited our language to be more precise, **replacing "formal methodology" with "principled methodology"**.
>
>
>
> ---
>
>
>
> > 2. In Shachi, the agent is decomposed into four components—Configs, Memory, Tools, and LLM. Since both retrieved memory content and tool outputs are provided to the LLM in textual form, potential conflicts or negative interference between these inputs may occur. As a formal methodology, how does Shachi ensure that such inconsistencies or interference are systematically prevented or mitigated at the architectural level?
>
>
> We thank the reviewer for pointing out a key design challenge for any sophisticated agent architecture. A core principle of Shachi is that the framework does not hard-code a single, universal conflict-resolution policy. Instead, our principled methodology is designed to provide researchers with the flexibility to implement and test different resolution strategies. Concretely, the conflict resolution can be handled by the LLM, which is guided by its components. For example, a researcher can use the Configs module to set an explicit policy saying that if memory of a past event conflicts with new information from a tool, prioritize the tool's output (as Appendix D.2 points out, an LLM is able to differentiate between information from its Memory and data from a Tool). This ability to systematically modify and test these resolution policies is a core strength of Shachi.

---

### Official Review · Reviewer_u6ZH · 2025-11-01

**Soundness:** 2
**Presentation:** 3
**Contribution:** 3
**Rating:** 6
**Confidence:** 3

**Summary:**

This paper presents Shachi, a principled and modular framework for large language model–based agent-based modeling (ABM). It standardizes agent design by decomposing each agent’s policy into four cognitive components: Configs, Memory, Tools, and an LLM reasoning engine, and provides a three-level benchmark suite for systematic evaluation. Experiments show that Shachi reproduces existing results, supports cross-task generalization, enables novel studies such as memory transfer and multi-world simulations, and achieves external validity by modeling real-world market reactions to a U.S. tariff shock.

**Strengths:**

Originality: The paper is original in proposing a unified and modular methodology for LLM-based agent-based modeling (ABM).  Rather than introducing a new agent or task, it formalizes how agents and environments should interact, filling an existing methodological gap in the field.

Quality: The experimental design is broad and novel, including real world task and new case studies such as memory transfer, multi-world interaction, and market simulations.  These diverse experiments support the framework’s robustness and generality.

Clarity: The paper shows a clear motivation and unifying perspective. The introduction, method and proposed framework are logically connected and easy to follow.

Significance: The work establishes a solid methodological foundation for future research on LLM-driven multi-agent systems. By standardizing interfaces and modularizing agent cognition, Shachi has the potential to become a reference framework for reproducible, scientific simulation studies using large language models.

**Weaknesses:**

1) Simplistic baselines and weak comparisons.
The baselines are mainly random agents or plain LLMs without modular components. That is too limited to fully substantiate Shachi’s methodological advantages.  More meaningful comparisons could include systems that also use LLM + Memory but lack Shachi’s standardized interface, or other existing LLM-based ABM frameworks reimplemented under the same conditions.  In addition, the current results omit statistical indicators such as variance, confidence intervals, and significance tests, making it difficult to evaluate robustness and reproducibility.  Incorporating richer baselines and proper statistical reporting would greatly strengthen the empirical evidence.

2) Presentation and clarity issues.
The paper’s organization is dense, with technical details scattered between sections and appendices. Some details, figures and tables could be better integrated with the main text, and clearer explanations of evaluation metrics and experimental settings would improve readability.

**Questions:**

1) How scalable is the Shachi framework in practice, including LLM calls, time scalability with increasing agent numbers, and space scalability?
2) Could the authors report variance, confidence intervals, or significance tests to substantiate the reliability of these results?
3) How sensitive are the reproduction outcomes to the choice of backend LLMs and prompts?

---

> ### Author Response · Authors · 2025-11-21
> **Responses to Reviewer u6ZH (Part 1)**
>
> We thank the reviewer for their constructive feedback and for highlighting Shachi's value as a unified and modular methodology that addresses a key methodological gap in the field. We are encouraged by your positive assessment of our broad experimental design, the framework's robustness, and the logical clarity of our motivation. We appreciate the opportunity to substantiate our claims with rigorous statistical evidence and scalability analysis.
>
> To address your concerns, we have made the following revisions and additional analyses:
> - Statistical Validation (W1, Q2): Clarified that our baselines represent complex, task-specific implementations and conducted rigorous Equivalence Testing (TOST) to prove Shachi’s reproducibility in Appendix C.1 and Section 4.1.
> - Improved Presentation (W2): Integrated key experimental settings into the main text and restructured Appendix B to provide a clearer mapping between tasks and details.
> - Scalability Analysis (Q1): Added empirical validation demonstrating time scalability up to 500 agents in Appendix C.6.
> - Sensitivity Analysis (Q3): Verified robustness against model and prompt variations using statistical testing.
>
> Please find our detailed responses below.
>
>
>
> ---
>
>
> Weaknesses:
> > 1, Simplistic baselines and weak comparisons. The baselines are mainly random agents or plain LLMs without modular components. That is too limited to fully substantiate Shachi’s methodological advantages. More meaningful comparisons could include systems that also use LLM + Memory but lack Shachi’s standardized interface, or other existing LLM-based ABM frameworks reimplemented under the same conditions. In addition, the current results omit statistical indicators such as variance, confidence intervals, and significance tests, making it difficult to evaluate robustness and reproducibility. Incorporating richer baselines and proper statistical reporting would greatly strengthen the empirical evidence.
>
>
> We thank the reviewer for their insightful feedback. We will first clarify the purpose of our baselines and then present our new, rigorous statistical validation.
>
> **The Purpose of Baselines and Table 1**
> The primary purpose of Table 1 was not to compare Shachi to weak baselines, but to demonstrate its reproducibility against the original, complex implementations. In the original submission, we used the "Shachi (Ours)" column to show the Mean Absolute Error (MAE) between the base implementation and our implementation. We must clarify that this "base implementation" is precisely the system the reviewer described: a system where the original benchmark designers custom-built their components (e.g., LLM, Memory) and which lacks Shachi’s standardized interface. Our goal is to prove that Shachi's standardized interface can faithfully reproduce the results of these existing benchmarks that were built without a unified interface.
>
> **Statistical Approach: Equivalence Testing (TOST)**
> To substantiate this claim of reproducibility with statistical rigor, a simple comparison to random baselines is insufficient. Instead, we must directly test whether the results from the original implementation and our Shachi implementation are statistically equivalent. For this purpose, we use the Two One-Sided Tests (TOST) procedure. Generally speaking, robust statistical tests, including TOST, require a larger number of samples for each condition. Running this analysis across all experiments is not feasible in LLM-based ABM, due to the high computational cost, both in terms of budget and time (e.g., the experiments below cost several thousand dollars).
>
> Therefore, we have selected one representative benchmark from each complexity level to perform this validation: Level 1 (EmergentAnalogies), Level 2 (AuctionArena), and Level 3 (Sotopia). Following standard practice, for all experiments, we performed a Welch TOST analysis with N=30 runs per condition, using a 90% confidence interval (α = 0.05 for each one-sided test).
>
>
> (Continued to the next post)

---

> ### Author Response · Authors · 2025-11-21
> **Responses to Reviewer u6ZH (Part 2)**
>
> (Continued from the previous post)
>
> **Level 1 - Emergent Analogies**
>
> For Level 1, we use Emergent Analogies, a task that requires an agent to infer the continuation of a matrix. The task is categorized by the number of digits to predict (from 'one' to 'five'). We set an equivalence boundary ($\Delta$) of 0.05. This means we can conclude the implementations are "equivalent" if we are 90% confident that the true difference between them is less than $\pm 0.05$. The results of the TOST analysis are as follows:
>
> |Problem Category|Base Mean ± std (N=30)|Shachi Mean ± std (N=30)|90% CI of difference|p(TOST)|Statistically Equivalent?|
> |---|---|---|---|---|---|
> |1-digit prediction|0.969 $\pm$ 0.015|0.977 $\pm$ 0.013|(0.001, 0.013)|$8.11\text{e-}18$|Yes (✓)|
> |2-digit prediction|0.816 $\pm$ 0.031|0.844 $\pm$ 0.032|(0.015, 0.042)|0.00521|Yes (✓)|
> |3-digit prediction|0.653 $\pm$ 0.024|0.664 $\pm$ 0.023|(0.000, 0.021)|$1.07\text{e-}08$|Yes (✓)|
> |4-digit prediction|0.512 $\pm$ 0.041|0.520 $\pm$ 0.039|(-0.009, 0.025)|$6.63\text{e-}05$|Yes (✓)|
> |5-digit prediction|0.391 $\pm$ 0.031|0.392 $\pm$ 0.030|(-0.012, 0.015)|$4.32\text{e-}08$|Yes (✓)|
>
> All metrics satisfied the equivalence criterion.
>
> **Level 2 - Auction Arena**
> For Level 2, we use Auction Arena, a task that evaluates agents' strategic planning and execution in a dynamic auction environment where they must manage resources, assess risks, and adapt their bidding strategies.
>
> In accordance with the original paper, we employ TrueSkill to quantify performance based on the final rankings of each session. We utilize the mean ($\mu$) of the estimated skill distribution as the metric for this analysis. We set an equivalence boundary ($\Delta$) of 2.0.
>
> |Metric|Base Mean $\pm$ std (N=30)|Shachi Mean $\pm$ std (N=30)|90% CI of Difference|p(TOST)|Statistically Equivalent?|
> |---|---|---|---|---|---|
> |TrueSkill $\mu$|27.191 $\pm$ 1.665|26.220 $\pm$ 2.067|(-1.781, -0.160)|0.019|Yes (✓)|
>
> All metrics satisfied the equivalence criterion.
>
> **Level 3 - Sotopia**
>
>
> We use the Sotopia task, which evaluates agent performance in complex social interactions across multiple dimensions. The task includes seven reward types that capture different aspects of the interaction, such as goal achievement, social adherence, believability, and financial outcomes. We set the equivalence boundary to Δ = 0.20.
>
> |Reward Type|Base Mean ± std (N=30)|Shachi Mean ± std (N=30)|90% CI of Difference|p(TOST)|Statistically Equivalent?|
> |---|---|---|---|---|---|
> |believability|9.690 ± 0.133|9.671 ± 0.123|(-0.0747, 0.0360)|5.29e-07|Yes (✓)|
> |relationship|3.551 ± 0.289|3.525 ± 0.189|(-0.1321, 0.0795)|0.00413|Yes (✓)|
> |knowledge|5.959 ± 0.458|5.956 ± 0.363|(-0.1810, 0.1756)|0.0347|Yes (✓)|
> |secret|-0.027 ± 0.043|-0.033 ± 0.054|(-0.0271, 0.0151)|3.82e-23|Yes (✓)|
> |social_rules|-0.035 ± 0.036|-0.030 ± 0.031|(-0.0087, 0.0200)|1.92e-30|Yes (✓)|
> |financial|0.701 ± 0.136|0.692 ± 0.140|(-0.0688, 0.0501)|7.53e-07|Yes (✓)|
> |goal|8.763 ± 0.239|8.715 ± 0.227|(-0.1485, 0.0525)|0.00709|Yes (✓)|
>
> All metrics satisfied the equivalence criterion.
>
> Across all three complexity levels, TOST analysis confirms that Shachi faithfully reproduces the original implementations with statistical rigor (all p < 0.05). These results demonstrate that Shachi's standardized interface achieves reproducibility without sacrificing fidelity to the original benchmarks. We **created Appendix C.1** to add the results and discussions.

---

> ### Author Response · Authors · 2025-11-21
> **Responses to Reviewer u6ZH (Part 3)**
>
> > 2, Presentation and clarity issues. The paper’s organization is dense, with technical details scattered between sections and appendices. Some details, figures and tables could be better integrated with the main text, and clearer explanations of evaluation metrics and experimental settings would improve readability.
>
> We appreciate the feedback regarding the organization and density of the paper. Due to strict initial page constraints, we were forced to move several key experimental details and discussions to the Appendix. In the revised manuscript, we have utilized the allowed additional page limit to significantly improve readability:
>
> 1. Integrated Key Details: We have moved the experimental settings for **Section 4.1** and the detailed discussion of results from the Appendix back into the main text to ensure a cohesive narrative.
> 2. Reorganized Appendices: We have **restructured Appendix B** to provide a clearer, one-to-one mapping between tasks and their respective experimental settings, figures, and tables, making it easier for readers to locate specific technical details.
>
> Please let us know if any parts of the presentation remain unclear or if further structural improvements are needed.
>
> ---
>
> Questions:
> > 1, How scalable is the Shachi framework in practice, including LLM calls, time scalability with increasing agent numbers, and space scalability?
>
> Our framework scales in both time and space.
>
> **Time scalability.** Because the framework itself performs little computation, the wall-clock time is dominated by LLM calls. To make this efficient, our framework uses Python’s async, so LLM calls are parallelized. Let a single LLM call take T time on average. With N agents, R rounds, and an LLM-call parallelism of P, the runtime is approximately TNR/P. There is no framework-level cap on P, and Python async is lightweight, so in practice P can be increased until the LLM API’s rate limits are reached. Ideally, we can take P \approx N, in which case the runtime becomes TR, i.e., theoretically independent of the number of agents N, though slight overheads from non-LLM processing may appear in practice. In many settings, T is a few seconds to a few minutes, and R is at most a few dozen.
>
> **Space scalability.** The framework and the environment/agent implementations maintain only the necessary information, so memory usage scales well. Even in the pessimistic case where the entire interaction history is retained, assuming the average size of a single LLM response is M, the total retained history is MNR. In typical cases, M is on the order of hundreds to thousands of tokens. Thus, space also scales linearly with the number of agents.
>
> **Empirical Validation.** To evaluate time scalability, we measured the execution time cost of the StockAgent environment using `gpt-5` while varying the agent population from 50 to 500. We recorded the time required for a complete simulation cycle comprising four sessions: loan decision, trading, next-day prediction, and reporting.
>
> |Number of Agents|Execution Time|
> |----------------|--------------|
> |50|7 min 53 secs|
> |100|7 min 59 secs|
> |500|9 min 9 secs|
>
> **Discussion.** The results strongly validate the efficiency of our parallel processing implementation. Consistent with our theoretical model, the execution time remained nearly constant regardless of the population size. Notably, scaling from the default 50 agents to 500 agents (a 10-fold increase) resulted in only a marginal increase in total execution time (from ~8 min to ~9 min). This demonstrates that the runtime is effectively independent of the agent count, with only minimal overheads attributed to non-parallelizable processing such as data aggregation.
>
> We added this result and **discussion in Appendix C.6.**
>
>
> ---
>
> > 2, Could the authors report variance, confidence intervals, or significance tests to substantiate the reliability of these results?
>
> As we discussed in the above TOST test, we reported variance, confidence intervals, and significance tests **in Appendix C.1**.

---

> ### Author Response · Authors · 2025-11-21
> **Responses to Reviewer u6ZH (Part 4)**
>
> > 3, How sensitive are the reproduction outcomes to the choice of backend LLMs and prompts?
>
> We thank the reviewer for this important question. To address this, we extended the TOST analysis to verify if statistical equivalence is maintained when changing backend LLMs and prompts. Due to the high computational cost of complex ABMs, we employed a two-tiered approach: rigorous testing ($N=30$) on the efficient EmergentAnalogies task, and trend verification ($N=3$) on the complex Sotopia task.
>
> ## Rigorous Sensitivity Analysis (EmergentAnalogies)
> We utilized the same condition TOST ( 90% confidence interval, α = 0.05 for each one-sided test). We tested if Base and Shachi remain statistically equivalent ($N=30, \Delta=0.05$) under identical modifications. The TOST results above confirmed their equivalence in the default setting with `gpt-4`. We further compared them using:
> - Prompt Change: Reordering questions within the prompt.
> - LLM Change: Using `gpt-4o-mini` with the default prompt.
>
>
> | Model | Prompt | 1-digit | 2-digit | 3-digit | 4-digit | 5-digit | Overall |
> |---|---|---|---|---|---|---|---|
> | `gpt-4` | Modified | Yes (✓) | Yes (✓) | Yes (✓) | Yes (✓) | Yes (✓) | **(5/5 Passed)** |
> | `gpt-4o-mini` | Default | Yes (✓) | Yes (✓) | Yes (✓) | Yes (✓) | Yes (✓) | **(5/5 Passed)** |
>
> The results confirm that Shachi maintains statistical equivalence across all configurations.
>
>
>
> ## Verification on Complex Tasks (Sotopia)
>
>
> For Sotopia, we verified performance alignment with $N=3$ runs using `gpt-4o-mini`. The TOST results above confirmed their equivalence in the default setting with `gpt-4o-mini`. We further validated robustness under two specific conditions:
> 1. Prompt Variation: with an additional "stubborn" instruction added to the prompt.
> 2. Model Variation: Using `gemini-2.5-flash` with the default prompt.
>
>
> | Model | Prompt | Metric | Base Mean ± std (N=3) | Shachi Mean ± std (N=3) |
> |---|---|---|---|---|
> | `gpt-4o-mini` | Modified | BEL | 9.620 ± 0.105 | 9.417 ± 0.065 |
> | | | REL | 2.800 ± 0.271 | 2.737 ± 0.145 |
> | | | KNO | 5.170 ± 0.227 | 4.987 ± 0.636 |
> | | | SEC | -0.067 ± 0.058 | -0.050 ± 0.050 |
> | | | SOC | -0.070 ± 0.104 | -0.070 ± 0.089 |
> | | | FIN | 0.477 ± 0.031 | 0.427 ± 0.045 |
> | | | GOAL | 8.083 ± 0.135 | 8.063 ± 0.263 |
> | `gemini-2.5-flash` | Default | BEL | 9.683 ± 0.040 | 9.653 ± 0.040 |
> | | | REL | 2.883 ± 0.284 | 2.863 ± 0.170 |
> | | | KNO | 5.530 ± 0.442 | 5.600 ± 0.645 |
> | | | SEC | -0.040 ± 0.069 | -0.033 ± 0.058 |
> | | | SOC | -0.047 ± 0.025 | -0.077 ± 0.032 |
> | | | FIN | 0.577 ± 0.116 | 0.517 ± 0.012 |
> | | | GOAL | 8.667 ± 0.021 | 8.677 ± 0.214 |
>
>
>
> Even with a smaller sample size, the means and standard deviations demonstrated consistent tendencies across different configurations. These results suggest that the qualitative patterns between Shachi and the Base model remain stable and are not sensitive to the choice of backend LLMs or prompts.

---

### Official Review · Reviewer_NkDa · 2025-11-03

**Soundness:** 3
**Presentation:** 3
**Contribution:** 3
**Rating:** 4
**Confidence:** 4

**Summary:**

This paper introduces Shachi, a formal methodology and modular framework for LLM-based agent-based modeling (ABM). The authors propose decomposing an agent's policy into four core components: Configuration (intrinsic traits), Memory (contextual persistence), Tools (extended capabilities), and an LLM reasoning engine. The framework is validated on a 10-task benchmark suite spanning three levels of social complexity (single-agent, non-communicative multi-agent, and communicative multi-agent).

**Strengths:**

1, Well-motivated problem: The paper addresses a genuine issue in LLM-based ABM research - the lack of standardized methodology leading to fragmented, irreproducible results. This is an important problem for the field.

2, Principled architecture design: The decomposition of agent policy into Configuration, Memory, Tools, and LLM is intuitive and grounded in cognitive science principles. The separation of agent architecture from environment through a standardized interface is a solid engineering contribution.

3, Strong empirical validation: The reproduction study (Table 1) shows low MAE across all tasks, demonstrating the framework's ability to faithfully replicate prior work. The external validity experiment with the U.S. tariff shock is particularly compelling.

4, Novel exploratory studies: The "carrying memory to next life" and "living in multiple worlds" experiments showcase creative applications enabled by the modular design, though they could benefit from deeper analysis.

**Weaknesses:**

1, Limited theoretical novelty: While the engineering contribution is solid, the conceptual decomposition (LLM + config + memory + tools) is relatively straightforward and has been implicitly used in prior agent frameworks. The paper doesn't provide strong theoretical justification for why this particular decomposition is optimal or complete.

2, Insufficient comparison with existing frameworks: The paper mentions AutoGen, Concordia, and EDSL but dismisses them too quickly without detailed empirical comparison. A systematic comparison showing Shachi's advantages would strengthen the contribution. The claim that these frameworks are "not designed for reproducible social simulation" needs more substantiation.

3, Cross-task generalization results are underwhelming: Table 2 shows that most agents perform similarly across tasks (scores ≈ 1.0), which undermines the claim about the importance of modular components. The only clear difference is when tools are missing for tool-requiring tasks, which is unsurprising.

4, Limited discussion of failure modes: The paper doesn't adequately discuss when the framework might fail or produce unrealistic behaviors. For instance, what happens when agents face novel situations not covered by their configuration or tools?

5, Missing important implementation details:

- How are conflicts between tools and memory resolved when both could apply?
- What are the computational costs of the framework compared to alternatives?
- How sensitive are results to LLM temperature, prompt variations, and other hyperparameters?
- The "two-stage parsing" strategy (Appendix D.4) seems like a workaround but its necessity and implications aren't discussed

**Questions:**

See weaknesses.

---

> ### Author Response · Authors · 2025-11-21
> **Responses to Reviewer NkDa (Part 1)**
>
> We thank the reviewer for their constructive feedback and for recognizing our work as a well-motivated solution with a principled architecture design. We are particularly encouraged that you found our empirical validation, specifically the U.S. tariff shock experiment, to be compelling. We appreciate the opportunity to strengthen the theoretical positioning and comparative analysis of our framework.
>
> To address your concerns, we have made the following revisions and additional analyses:
>
> - Clarified Theoretical Novelty (W1): Articulated the cognitive science grounding of our modular decomposition and its necessity for reproducible research.
> - Expanded Comparisons (W2): Added a concrete comparison with EDSL and clarified the structural differences from AutoGen/Concordia in Appendix E.3.
> - Analyzed Granular Performance (W3): Provided a detailed breakdown of memory effects and specific metrics (e.g., Sotopia) to demonstrate the critical impact of modular components in Section 4.1 and Appendix C.2.
> - Discussed Failure Modes (W4): Added a discussion on agent behavior in out-of-distribution (OOD) scenarios to the Limitations section.
> - Provided Implementation Details (W5): Clarified conflict resolution strategy, conducted computational cost evaluation, sensitivity analysis (temperature/prompt/model) in Appendix C.4, and the rationale for the two-stage parsing strategy in Appendix D.4.
>
> Please find our detailed responses below.
>
> ---
>
> > 1, Limited theoretical novelty: While the engineering contribution is solid, the conceptual decomposition (LLM + config + memory + tools) is relatively straightforward and has been implicitly used in prior agent frameworks. The paper doesn't provide strong theoretical justification for why this particular decomposition is optimal or complete.
>
> We thank the reviewer for their feedback and acknowledge our solid engineering contribution. Our central contribution is a principled methodology and modular framework designed to solve the "methodological fragmentation" that currently hinders reproducible and cumulative research in LLM-based ABM. Our method's novelty is detailed as follows:
> 1. **Theoretical justification**: Our decomposition is not ad-hoc, it is explicitly grounded in established cognitive science principles. Specifically, we map Configs to psychological traits, Memory to working and episodic memory, Tools to cognitive augmentation, and cited papers for each of these in Section 3.2.
> 2. **From "implicit" to "explicit"**: The reviewer insightfully notes that these components have been implicitly used. This highlights the exact problem Shachi addresses. In prior works, these were implemented as one-off features in ad-hoc designs, preventing systematic analysis. Shachi, on the other hand, formalizes this decomposition into an explicit and modular architecture, enabling systematic analysis and flexible extensions.
> 3. **Empirical justification**: While we agree that claiming any single architecture is "optimal or complete" is difficult, Shachi provides strong empirical justification for this 4-part design. Section 4.2.3 (US tariff shock) serves as a direct cumulative ablation study. The key finding is that only the full cognitive architecture produced agent behaviors that aligned with real-world observations. This empirically demonstrates that our decomposition is necessary for establishing the external validity of agents. By open-sourcing Shachi, we provide the foundation for future researchers to systematically test and extend this architecture.

---

> ### Author Response · Authors · 2025-11-21
> **Responses to Reviewer NkDa (Part 2)**
>
> > 2, Insufficient comparison with existing frameworks: The paper mentions AutoGen, Concordia, and EDSL but dismisses them too quickly without detailed empirical comparison. A systematic comparison showing Shachi's advantages would strengthen the contribution. The claim that these frameworks are "not designed for reproducible social simulation" needs more substantiation.
>
> Thank you for pointing out the need for a clearer comparison with AutoGen, Concordia, and EDSL. Our original discussion was brief, so we appreciate the opportunity to clarify.
>
> These frameworks are well-designed and provide useful components such as flexible agents and practical building blocks that many users rely on. Their strengths are clear: they offer highly reusable building blocks for developers in conversational task automation or game-master-led interactions. At the same time, they are built with different goals in mind: they do not offer the structured agent and environment scaffolding (for example, standardized environments, orchestration, and reproducible execution settings) that our work focuses on. As a result, using them for social simulation typically requires researchers to build significant custom infrastructure, which is different from Shachi's integrated approach.
>
> To ground the comparison a bit more, we adapted EDSL to run a small subset of Level-1 PsychoBench tasks (the Empathy Questionnaire). With some effort, we were able to get results similar to ours. For the `gpt-4o-mini` model, both Shachi and EDSL produced empathy scores that were statistically higher than the human average using Welch's t-test. While doing this, we also confirmed that running the more complex Level-2 and Level-3 tasks would require much more engineering work than we could take on in this paper. This helps explain why deeper comparisons are challenging.
>
> We have updated the **Related Work section and Appendix E.3** to make these points clearer. Thanks again for the helpful suggestion.
>
> ---
>
> > 3, Cross-task generalization results are underwhelming: Table 2 shows that most agents perform similarly across tasks (scores ≈ 1.0), which undermines the claim about the importance of modular components. The only clear difference is when tools are missing for tool-requiring tasks, which is unsurprising.
>
> We appreciate the observation regarding Table 2. We clarify the specific impact of the modular components from the following perspectives:
>
> 1. **Importance of Memory in Non-Tool Contexts**: While the reviewer noted that differences seem limited to tool-requiring tasks, our results on AuctionArena suggest otherwise. AuctionArena does not require external tools, yet the memory-less EmergentAnalogies agent scores significantly lower (0.62) compared to memory-equipped agents (~1.0). This demonstrates that for complex tasks, the memory module is a critical performance factor, independent of tool availability.
> 2. **Qualitative Shift in Sotopia**: We recognize that the normalization in our main paper may have obscured fine-grained differences. However, examining Sotopia’s seven metrics individually reveals distinct agent profiles.
>     - **Greed vs. Goals**: The memory-less agent (EmergentAnalogies) scores highest on Financial (1.159) but low on Goal Completion (0.970). This suggests that without memory, the agent defaults to local optimization (maximizing immediate gain). In contrast, agents with long-term memory (AuctionArena; window 20, 10k token limit) achieve higher Goal Completion (1.005), successfully trading off immediate gains for strategic objectives.
>     - **Inverse Correlation in Relationship**: Interestingly, Relationship scores improve as memory capacity decreases: Sotopia (all history, 16k limit) < AuctionArena (window 20, 10k limit) < StockAgent (window 3). Specifically, StockAgent achieves a +7% higher score than Sotopia. This suggests that retaining full history constrains the agent to rigid task consistency, whereas limiting memory allows the agent to prioritize immediate social responsiveness over long-term coherence.
>
> These results demonstrate that both the availability and specific configuration of modules critically impact task performance. We have incorporated the analysis of memory effects (Point 1) and added the detailed Sotopia results and discussion (Point 2) to **Section 4.1 and Appendix C.2**.
>
> ---
>
> > 4, Limited discussion of failure modes: The paper doesn't ... agents face novel situations not covered by their configuration or tools?
>
> We thank the reviewer for this insightful comment, which highlights a critical research frontier for all ABM. Although our current limitations section in Sec 5 focuses on the challenges of environmental fidelity, the reviewer is correct that we did not explicitly detail agent behavior in unforeseen, OOD scenarios. **We have included this point in the limitations section**, noting that Shachi's modularity now provides a foundation to systematically study how agents adapt to such unforeseen scenarios.

---

> ### Author Response · Authors · 2025-11-21
> **Responses to Reviewer NkDa (Part 3)**
>
> > 5, Missing important implementation details:
> > - How are conflicts between tools and memory resolved when both could apply?
>
> We thank the reviewer for pointing out a key design challenge for any sophisticated agent architecture. A core principle of Shachi is that the framework does not hard-code a single, universal conflict-resolution policy. Instead, Shachi's modular architecture is an open-source foundation designed to provide researchers with the flexibility to implement and test different strategies. Concretely, the conflict resolution can be handled by the LLM, which is guided by its components. For example, a researcher can use the Configs module to set an explicit policy saying that if memory of a past event conflicts with new information from a tool, prioritize the tool's output (as Appendix D.2 points out, an LLM is able to differentiate between information from its Memory and data from a Tool). This ability to systematically modify and test these resolution policies is a core strength of Shachi.
>
>
>
> ---
>
>
>
> > - What are the computational costs of the framework compared to alternatives?
>
>
> We compared the computational costs (execution time) of the proposed framework, Shachi, against the existing Base implementation. The evaluation measured the execution time for one task chosen from each level: EmotionBench (Level 1), StockAgent (Level 2), and OASIS (Level 3). All experiments were done with `gpt-4o`.
>
>
> |Task|Base Time (m)|Shachi Time (m)|Time Ratio (Shachi/Base)|Notes|
> |---|---|---|---|---|
> |EmotionBench (level 1)|45.32|1.52|0.034|353 questions|
> |StockAgent (level 2)|20.41|14.28|0.700|3 days|
> |OASIS (level 3)|1.73|2.14|1.237|3 iterations|
>
>
> For StockAgent (Level 2) and OASIS (Level 3), Shachi achieved comparable performance, indicating that utilizing Shachi generally does not significantly alter the computational cost. Conversely, for EmotionBench (Level 1), where the Base implementation lacked parallel processing, Shachi employed async parallelism to achieve a dramatic speedup, reducing the execution time by approximately 30 times (Time Ratio 0.034). In conclusion, Shachi ensures no degradation in speed in our experiments while offering substantially higher computational efficiency for tasks where parallelism is lacking.

---

> ### Author Response · Authors · 2025-11-21
> **Responses to Reviewer NkDa (Part 4)**
>
> > - How sensitive are results to LLM temperature, prompt variations, and other hyperparameters?
>
>
> We thank the reviewer for raising the critical point about sensitivity to hyperparameters like temperature and prompts. In LLM-based Agent-Based Models (ABM), results are inherently dynamic; the accumulation of agent decisions means that small changes in hyperparameters can indeed alter the final outcome. Similarly, changing the backend LLM (e.g., `gpt-3.5` vs. `gpt-4`) is expected to reflect that model's specific characteristics in the results. This is a universal feature of LLM-based ABM, not a specific issue with Shachi.
>
> Therefore, we conduct two complementary analyses: (1) a general sensitivity evaluation to understand how hyperparameter changes affect system behavior across different configurations, and (2) a rigorous statistical test to verify that Base and Shachi implementations remain equivalent when subjected to identical modifications.
>
> ## General Sensitivity Evaluation
>
> We conducted sensitivity analysis on Level 3 Sotopia tasks by varying temperature, backend LLMs, and prompts (specifically, adding a "stubbornness" instruction to the system prompt).
>
> |Row|LLM|Temperature|Prompt|Believability|Relationship|Knowledge|Secret|Social Rules|Financial Benefits|Goal|
> |---|---|---|---|---|---|---|---|---|---|---|
> |1|`gpt-4o-mini`|0.5|-|9.693 ± 0.129|3.360 ± 0.619|5.597 ± 0.670|-0.003 ± 0.006|-0.047 ± 0.046|0.660 ± 0.151|8.623 ± 0.366|
> |2|`gpt-4o-mini`|0.5|added|9.620 ± 0.105|2.800 ± 0.271|5.170 ± 0.227|-0.067 ± 0.058|-0.070 ± 0.104|0.477 ± 0.031|8.083 ± 0.135|
> |3|`gpt-4o-mini`|1.0|-|9.543 ± 0.204|3.540 ± 0.581|5.830 ± 0.976|-0.067 ± 0.058|-0.063 ± 0.012|0.723 ± 0.021|8.603 ± 0.394|
> |4|`gemini-2.5-flash`|0.5|-|9.683 ± 0.040|2.883 ± 0.284|5.530 ± 0.442|-0.040 ± 0.069|-0.047 ± 0.025|0.577 ± 0.116|8.667 ± 0.021|
>
>
> Summary of Findings
>
> **Prompt (Row 1 and 2):** Adding the "stubborn" trait resulted in a notable decrease in the Relationship score ($3.360 \rightarrow 2.800$). This confirms that the agents successfully internalized the personality directive, leading to more socially difficult interactions as intended.
>
> **Temperature (Row 1 and 3):** Compared to prompt variations, changing the temperature (0.5 vs. 1.0) showed a limited impact on the metrics, suggesting the system maintains reasonable stability against sampling noise within this range in this Sotopia task.
>
> **Model (Row 1 and 4):** When changing the backend model to `gemini-2.5-flash`, we observed a decrease in both Relationship ($3.360 \rightarrow 2.883$) and Financial Benefits ($0.660 \rightarrow 0.577$). This variance illustrates that the simulation results appropriately reflect the inherent behavioral tendencies and alignment differences of the underlying models.
>
> We have **created a new Appendix C.4** to add the results and discussions.
>
>
>
> ## Rigorous Reproduction Sensitivity Analysis (EmergentAnalogies)
> Beyond general sensitivity patterns, we verify that Shachi maintains statistical equivalence with the Base implementation under identical modifications. We use the Two One-Sided Tests (TOST) procedure with $N=30$ and equivalence margin $\Delta=0.05$. TOST is the standard method for equivalence testing. Following standard practice, we conduct one-sided tests in both directions with α=0.05, yielding a 90% confidence interval.
>
> Robust statistical testing requires large sample sizes, and running repeated trials across all conditions is computationally prohibitive in LLM-based ABM due to both time and budget constraints. Therefore, we focus on the relatively simple and cost-efficient EmergentAnalogies benchmark for this rigorous validation. We tested if Base and Shachi remain statistically equivalent ($N=30$, $\Delta=0.05$) under identical modifications by varying temperature, prompt, and backend LLM:
> - Dafault setting: model `gpt-4`, temp 0, and base prompt.
> - Temperature setting: temp 0 -> 0.5
> - Prompt Change: Reordering questions within the prompt.
> - LLM Change: Using `gpt-4o-mini` with the default prompt.
>
>
> | Model | Temp | Prompt | 1-digit | 2-digit | 3-digit | 4-digit | 5-digit | Overall |
> |---|---|---|---|---|---|---|---|---|
> | `gpt-4` | 0.0 | Default | Yes (✓) | Yes (✓) | Yes (✓) | Yes (✓) | Yes (✓) | **(5/5 Passed)** |
> | `gpt-4` | 0.5 | Default | Yes (✓) | Yes (✓) | Yes (✓) | Yes (✓) | Yes (✓) | **(5/5 Passed)** |
> | `gpt-4` | 0.0 | Modified | Yes (✓) | Yes (✓) | Yes (✓) | Yes (✓) | Yes (✓) | **(5/5 Passed)** |
> | `gpt-4o-mini` | 0.0 | Default | Yes (✓) | Yes (✓) | Yes (✓) | Yes (✓) | Yes (✓) | **(5/5 Passed)** |
>
> The results confirm that Shachi maintains statistical equivalence with the original Base implementation across all tested configurations, demonstrating that our standardization preserves the original system's performance without introducing systematic deviations.

---

> ### Author Response · Authors · 2025-11-21
> **Responses to Reviewer NkDa (Part 5)**
>
> > - The "two-stage parsing" strategy (Appendix D.4) seems like a workaround but its necessity and implications aren't discussed
>
> We clarify that this is not a workaround, but a principled design choice. Recent work [1] shows that strictly enforcing rigid formats like JSON mode can degrade an LLM's reasoning performance. Our two-stage strategy is designed to mitigate this known trade-off. We decouple reasoning from formatting: the LLM first generates its reasoning in a less constrained manner (preserving performance), and we then robustly parse this output into the required structure (ensuring reliability). We **have updated Appendix D.4** to clarify this rationale and its implications.
>
> Reference
> [1] Zhi Rui Tam, Cheng-Kuang Wu, Yi-Lin Tsai, Chieh-Yen Lin, Hung-yi Lee, and Yun-Nung Chen. Let me speak freely? a study on the impact of format restrictions on large language model performance. In Proceedings of the 2024 Conference on Empirical Methods in Natural Language Processing: Industry Track, pp. 1218–1236, 2024.

---

### Author Response · Authors · 2025-11-21
**Message to all reviewers and AC**

We thank the reviewers for acknowledging our work as a well-motivated solution with a principled architecture design and a novel modular framework that addresses a key gap in the field. We are particularly encouraged that you found our empirical validation to be compelling and our 10-task benchmark to be a much-needed standardized testbed with logical clarity.

We highly appreciate your reviews that allowed us to clarify ambiguities and further improve the paper. **We have revised our paper to reflect these clarifications and improvements, with all changes highlighted in blue for your convenience.**

As a summary of our responses:

**Writing and Presentation**:
- Clarified the cognitive science grounding of our architecture (see response to reviewers NkDa and zhc3).
- Refined theoretical terminology to "principled methodology." (see Abstract and Introduction)
- Integrated key experimental settings into the main text for better flow (see Section 4.1).
- Restructured Appendix B to provide a clearer mapping between tasks and details (see Appendix B).
- Added discussion on failure modes in out-of-distribution (OOD) scenarios (see Section 5).
- Explained component conflict resolution strategies (see response to reviewers NkDa and zhc3).
- Clarified the rationale for the two-stage parsing strategy (see Appendix D.4).
- Articulated external validity goals regarding configurability (see response to reviewer zhc3).
- Incorporated suggested references on LLM agent methodology (see Section 3.1).

**Experiments and Analysis**:
- Conducted rigorous Equivalence Testing (TOST) to demonstrate statistical reproducibility (see Appendix C.1 and Section 4.1).
- Analyzed granular performance in Sotopia to demonstrate the critical role of modular components in cross-task generalization (see Section 4.1 and Appendix C.2).
- Conducted sensitivity analyses regarding temperature, prompt, and model variations (see Appendix C.4).
- Verified robustness of reproduction results against model and prompt variations using statistical equivalence testing (see response to reviewers NkDa and u6ZH).
- Demonstrated time scalability up to 500 agents in StockAgent, 10x the default setting (see Appendix C.6).
- Provided benchmarks comparing execution time against baselines (see response to reviewer NkDa).
- Clarified structural differences from existing frameworks and added concrete experimental comparisons with EDSL (see Section 2 and Appendix E.3).

We hope that our responses and revisions satisfactorily address the reviewers' concerns. If so, we kindly ask the reviewers to consider re-evaluating their ratings in light of these improvements.

---

### Meta-Review · Area_Chair_gf4W · 2026-01-02

**Summary:**

This paper proposes a modular LLM framework for LLM-based agent-based modeling. Two reviewers gave negative scores with concerns including limited adaptability to real-world applications and limited justification with GPT-4o-mini. Some concerns have been partially resolved during the rebuttal such as cognitive science grounding. Although the authors provided experiments to validate the performance such as cross-task generalization during the rebuttal, the paper is still lower than the threshold of ICLR.

**Reviewer Concerns:**

See above.

**Reviewer Scores:**

They tend to keep the scores.

---

### Decision · Program_Chairs · 2026-01-26

Reject